# ACTIVE REFINEMENT OF WEAKLY SUPERVISED MODELS

## ABSTRACT

Supervised machine learning (ML) has fueled major advances in several domains such as health, education and governance. However, most modern ML methods rely on vast quantities of point-by-point hand-labeled training data. In domains such as clinical research, where data collection and its careful characterization is particularly expensive and tedious, this reliance on pointillistically labeled data is one of the biggest roadblocks to the adoption of modern data-hungry ML algorithms. Data programming, a framework for learning from weak supervision, attempts to overcome this bottleneck by generating probabilistic training labels from simple yet imperfect heuristics (or labelling functions) obtained a priori from domain experts. We present WARM, Active Refinement of Weakly Supervised Models, a principled approach to iterative and interactive improvement of weakly supervised models via active learning. WARM directs domain experts' attention on a few selected data points that, when annotated, would most improve the label model's probabilistic accuracy. Gradient updates are then backpropagated to iteratively update the parameters of the individual expert labelling functions in the weak supervision model. Experiments on multiple real-world medical classification datasets reveal that WARM can substantially improve the accuracy of probabilistic labels used to train downstream classifiers, with as few as 30 queries to experts. Additional experiments with domain shift and artificial noise demonstrate WARM's ability to adapt to changing population characteristics as well as noisy initial labelling functions from the experts. These capabilities make WARM a potentially useful tool for deploying, maintaining, and auditing weakly supervised systems in practice.

## 1 INTRODUCTION

Machine learning (ML) has seen widespread adoption in several domains such as health, education and public policy. In the clinical context, supervised ML algorithms have been shown to be effective solutions to a gamut of problems, ranging from detecting abnormal heart rhythms in electrocardiogram (Goswami et al., 2022; Hannun et al., 2019), prognosticating neurological recovery using continuous electroencephalogram (Elmer et al., 2016; 2020) to detecting lung cancer in CT scans (Gao et al., 2019). However, most modern ML algorithms, especially large deep neural networks, rely on vast quantities of pointillistically labeled training data. In our experience, while raw clinical data is abundant, its careful annotation is not. Manually labeling large quantities of clinical data is often tedious, prohibitively expensive, prone to error, and acutely unscalable.

Recently, many studies have explored the use of cheap, potentially noisy sources of supervision to noisily annotate large training sets (Goswami et al., 2022; Fries et al., 2019; Saab et al., 2020) using weak supervision. These methods combine these sources to create noisy labels on unseen data, which can then be used to learn powerful discriminative end models. In principle, these methods resemble other *repeated-labeling* techniques such as crowd-sourcing (Li et al., 2013), or distant supervision relying on external knowledge bases (De Sa et al., 2016). In this paper, we leverage data programming (Ratner et al., 2016) which uses a generative model to integrate expert knowledge expressed in the form of noisy heuristics (*labeling functions* or LFs) to probabilistically label training data. Since the quality of the training labels is strongly correlated with that of labeling functions, unrefined weak supervision may not always yield optimal performance. Hence, in practice, experts

using data programming often spend several hours iteratively designing new heuristics or refining existing ones (Varma & Ré, 2018). Our work aims to streamline and reduce this effort.

Another closely related issue is that of domain shift (also referred to as *knowledge shift*). Imbalance in characteristics of source and target populations may render ML models trained on one domain perform poorly on another. This domain shift, especially in the form of population characteristics like gender or race, is prevalent in clinical settings (Thiagarajan et al., 2018). In the context of weak supervision, we are interested in the related problem of knowledge shift. Specifically, we want to train effective ML classifiers while tailoring knowledge of the source population to the target population (e.g. heart disease in young and old patients) via small changes in the LFs.

To this end, we propose WARM (Active Refinement of Weakly Supervised Models), a principled approach towards quick, iterative, and interactive refinement of weakly supervised models[1]. WARM directs expert attention to specific data points in the training set, which if labeled, would improve the labeling functions and the resulting label model the most. Specifically, at each iteration WARM directs an expert to label the most uncertain data point, and then updates decision parameters of the labeling functions to reduce the uncertainty.

Experiments on multiple real-world medical classification datasets of varying sizes and complexity, reveal that with as few as 30 queries to an expert, WARM can significantly improve label model accuracy resulting in higher quality training labels. Moreover, WARM outperforms existing approaches to active weak supervision (Biegel et al., 2021; Nashaat et al., 2018) and supports interpretability by allowing experts to track changes in LFs. Moreover, auxiliary experiments on a subset of our datasets show that WARM can not only combat knowledge shift but also de-noise LFs. For example, we found that WARM could adapt LFs to detect the presence of heart disease defined on a young population of patients, to an older population, resulting in a significant $1 - 3\%$ increase in label and end model performance. We also found that WARM was able to easily refine noisy LFs to achieve label and end model performance on par with no noise.

## 2 RELATED WORK

There is a small but growing body of work on active strategies for weak supervision. A few studies combine data programming with active learning on data points. For instance, Nashaat et al. (2018) proposed a method which actively obtained expert-labeled data via uncertainty sampling and corrected LF votes on individual data points to reflect the true labels. While their method improved the probabilistic predictions, it did not change the label model per se. Biegel et al. (2021) recently proposed Active WeaSuL which uses experts labels to learn how to best combine LFs. Specifically, they add a penalty term to the data programming loss function to nudge its optimal parameters to a configuration where the label model predictions agree with the expert labels. They also proposed a novel active sampling strategy tailored to the problem, which splits the data into buckets and measures the KL-divergence between the expert and estimated probabilistic labels to query new labels.

Some other studies have instead used labeled data to inspire ideas for new labeling functions (Cohen-Wang et al., 2019), and yet others have used them to automatically come up with new one (Awasthi et al., 2020; Boecking et al., 2020). Cohen-Wang et al. (2019) for instance proposed two simple strategies (*i.e.* data points where LFs abstain or disagree the most) to iteratively select subsets of data to inspire domain experts to create new LFs. On the other hand, Boecking et al. (2020) introduced an interactive weak supervision method that automatically improves a set of LFs based on feedback from users.

Alternatively, to reduce human effort to an absolute minimum, some studies have attempted to automate weak supervision by automatically generating heuristics based on a small labeled subset of the data (Varma & Ré, 2018; Nashaat et al., 2020). For instance, Varma & Ré (2018) proposed Snuba which automatically trains decision trees, logistic regression models and k-nearest neighbor classifiers as LFs, tuned for accuracy and coverage (amount of data that the LF predicts a class as oppose to abstaining from predicting any class), and later pruned to ensure sufficient dissimilarity. Finally, Nashaat et al. (2020) proposed Asterisk, extending their prior work on active weak supervision

---

[1]We will release the source code for WARM at www.github.com/anonymized_for_review after peer-review.

by now automatically generating heuristics from a small labeled dataset. In our experience, these methods are highly dependent on the quality and size of the labeled validation set. Furthermore, they may require large diverse labeled datasets for complex prediction tasks as evidenced by our experiments on the *diabetes dataset*. Additionally, `Asterisk` needs a large amount (up to $\sim 7\%$ of training data) of actively labeled data points for good performance.

# 3 METHODOLOGY

## 3.1 PROBLEM FORMULATION AND MODELING ASSUMPTIONS

Let $(\mathbf{x}, y) \sim \mathcal{D}$ denote the data generating distribution, where each data point $\mathbf{x} \in \mathbb{R}^d$ is associated with a latent class label $y \in \{1, \ldots, k\}$. Users provide an unlabeled training dataset $\mathcal{S}_u = \{(\mathbf{x}_i, y_i)\}_{i=1}^n$, where $y_i$ is unobserved, and $m$ *soft unipolar* labeling functions (LFs) $\Gamma$ which we define as follows.

Since we only consider LFs that predict a single class, let $pol(f) \in \{1, \ldots, k\}$ define the polarity (or the single class that it predicts) of a LF $f$. We define $\Gamma = \{\gamma_{\tau,j}\}_{j=1}^m$, $\gamma_{\tau,j} : \mathbb{R}^d \to [0, 1]$, such that $\gamma_{\tau,j}(\mathbf{x}_i, c) \triangleq \mathbb{1}_{\{c = pol(\gamma_{\tau,j})\}} \hat{p}(y_i = c \mid \mathbf{x}_i)$, where $\hat{p}$ is any arbitrary scoring function differentiable with respect to its parameters $\tau$.

Specifically, each LF $\gamma_{\tau,j}$ depends on one or more parameters in the parameter vector $\tau$ and expresses the likelihood that a data point $\mathbf{x}_i$ belongs to a particular class as identified by the polarity function $pol(\gamma_{\tau,j})$. We drop the subscript $\tau$ in the rest of our discussion for the sake of brevity.

## 3.2 MOST COMMON CLASSES OF LABELING FUNCTIONS ARE DIFFERENTIABLE

Fig. 1(ii) is an example of an expert-defined LF to characterize normal clinical findings in electroencephalogram (EEG) recordings[2] which depends on the `high_baseline` parameter.

Our methods assume that all the LFs in the label model are differentiable with respect to their decision parameters. Many frequently used classes of LFs are intrinsically differentiable with respect to their parameters, such as multi-layer perceptrons, convolutional neural networks, log-linear models, neural language models (Hooper et al., 2020; Varma & Ré, 2018). Other common classes of LFs can be easily transformed into their soft counterparts. One example is decision stumps (IF <CONDITION> THEN CLASS  ELSE CLASS <J>) (Varma & Ré, 2018). More generally, a AND or OR could also be made differentiable by fuzzy logic. To transform crisp decision stumps into their soft equivalents, we can follow prior work on decision trees (Suárez & Lutsko, 1999). Specifically, consider a heuristic of the form: $\lambda \triangleq \mathbf{d}^\intercal \mathbf{x} > a$, where $\mathbf{d}^\intercal \mathbf{x}$ defines a new composite variable and $a \in \tau$ is a decision parameter. Then we define its fuzzy equivalent as $\gamma \triangleq \text{sigmoid}(\mathbf{d}^\intercal \mathbf{x} - a)$, where $\text{sigmoid}(\mathbf{x}) = \frac{1}{1+\exp(-\mathbf{x})}$. Intuitively, while $\lambda$ indicates which side of the hyperplane $\mathbf{d}^\intercal \mathbf{x} = a$ a point lies, $\gamma$ encapsulates a probabilistic notion of distance from the decision boundary. Using the corresponding

```python
def high_aEEG_baseline_NORMAL(x, parameters):
    """High shaggy aEEG baseline constantly at an
    amplitude of around 10-20 mV, then NORMAL EEG.
    """
    # x[:, 9] : Average EEG baseline
    votes = np.where(
        x[:, 9] >= self.parameters['high_baseline'], 1, 0)
    return votes
```
(i)
```python
class high_aEEG_baseline_NORMAL(torch.nn.Module):
    """High shaggy aEEG baseline constantly at an
    amplitude of around 10-20 mV, then NORMAL EEG.
    """
    def __init__(self, parameters):
        super().__init__()
        self.parameters = parameters

    def forward(self, x):
        # x[:, 9] : Average EEG baseline
        votes = torch.sigmoid(
            x[:, 9] - self.parameters['high_baseline'])
        return votes
```
(ii)

Figure 1: Examples of (i) crisp and (ii) soft equivalents of a labeling function to characterize normal clinical findings in EEG data. While both versions make identical predictions, the latter is differentiable with respect to its decision parameter, `high_baseline`.

---

[2]Specifically, the LF is developed to work with amplitude-integrated EEG, which is a widely used quantitative summary of raw multi-channel EEG data.

Zadeh operators, conjunctions are converted via $(\text{AND}(x,y) \rightarrow \text{MIN}(x,y))$, disjunctions are converted via $(\text{OR}(x,y) \rightarrow \text{MAX}(x,y))$ and negations are converted via $(\text{NOT}(x) \rightarrow (1-x))$ (Zadeh, 1996). There are still other ways to obtain soft equivalents of crisp LFs, for instance by training feed-forward neural networks to approximate classes of functions under reasonable assumptions on their continuity and topology.

This extension of crisp LFs into their soft counterparts not only enables them to express confidence on their predictions, but also better capture fuzzy decision boundaries, common in most practical settings. Additionally, restricting LFs to be unipolar can be done without any loss of generality, since complex LFs can be easily broken down into their unipolar components. Moreover, in our experience, several simple unipolar LFs tend to perform better than a few complex multipolar LFs in practice. Each soft LF $\gamma_j$ can be converted to its *hard* equivalent $\lambda_j : \mathbb{R}^d \rightarrow \{0, pol(\lambda_j)\}$ using a threshold function, where $0$ means that the LF abstained from voting for class $pol(\lambda_j)$. In particular, we have that

$$\lambda_j = \begin{cases} c, & \text{if } \gamma_{\tau,j}(x,c) \geq 0.5 \\ 0, & \text{otherwise} \end{cases}$$

### 3.3 Probabilistic Label Estimation

Given the unlabeled training dataset $\mathcal{S}_u$ and an initial set of LFs denoted by $\Gamma^{(1)}$, our goal is to efficiently improve the accuracy of LFs by refining their parameters $\tau$ based on expert feedback, and ultimately assign probabilistic labels $\hat{p}(y \mid \Gamma^{(t)}, \mathbf{x})$ to the training data. Here $\Gamma^{(t)}$ denotes the set of LFs after $t^{th}$ iteration of WARM. The probabilistic labels are used to further train a downstream classifier $f : \mathcal{X} \rightarrow \mathcal{Y}$. We refine the LFs automatically by asking domain experts to provide labels for a few highly uncertain data points, and back propagating LF errors to update their parameters $\tau$.

In order to estimate probabilistic labels $p(y \mid \Gamma^{(t)}, \mathbf{x})$, we propose the following weighted majority-vote label model:

$$\hat{\mathbf{p}}_{il} = \frac{\mathbf{b}_l + \sum_{j=1}^{m} \gamma_j^{(t)}(\mathbf{x}_i, l)\hat{\theta}_j^{(t)}}{\sum_{c=1}^{k}(\mathbf{b}_c + \sum_{j=1}^{m} \gamma_j^{(t)}(\mathbf{x}_i, c)\hat{\theta}_j^{(t)})} \tag{1}$$

such that $\hat{\mathbf{p}}_{il} = p(y_i = l \mid \Gamma^{(t)}, \mathbf{x}_i)$ is the estimated probability that $\mathbf{x}_i$ belongs to class $l$, while $\mathbf{b}_l = p(y = l)$ denotes its prevalence. $\gamma_j^{(t)}(\mathbf{x}_i)$ is the probabilistic vote of $j^{th}$ LF and $\hat{\theta}_j^{(t)}$ is its unobserved accuracy, estimated using the data programming (DP) label model as defined in Eq. 2 (Ratner et al., 2016).

We ensure that $\hat{\mathbf{p}}_{ik}$ is a probability by passing $\hat{\mathbf{p}}_i$ through a softmax function, and note that our model is differentiable with respect to all LF parameters $\tau$. The DP label model is not differentiable with respect to LF parameters, and therefore cannot be directly used to estimate probabilistic labels while still having the ability to update LF parameters. However, data programming has been shown to perform well in several weak supervision applications and has some theoretical guarantees for estimating the latent accuracy of LFs by optionally modeling dependencies between them. Thus, in order to iteratively improve the accuracy of LFs and estimate probabilistic labels, we alternate between two objectives at the $t^{th}$ time step:

**Estimate the accuracy of LFs keeping their parameters $\tau$ fixed.** Let $[\Lambda^{(t)}]^{n \times m} = \{0, 1, \ldots, k\}^{n \times m}$ denote the observed matrix of crisp LF outputs at the $t^{th}$ time step, such that $[\Lambda^{(t)}]_{ij} = \lambda_j^{(t)}(\mathbf{x}_i)$ is the thresholded output of LF $\gamma_j$ on $\mathbf{x}_i$. We define factors for LF accuracy and propensity as $\phi^{Acc}([\Lambda^{(t)}]_{ij}, y_i) \triangleq \mathbb{1}_{\{[\Lambda^{(t)}]_{ij} = y_i\}}$ and $\phi^{Lab}([\Lambda^{(t)}]_{ij}, y_i) \triangleq \mathbb{1}_{\{[\Lambda^{(t)}]_{ij} \neq 0\}}$, respectively. We then follow Ratner et al. (2016) to define the joint distribution of the latent class label $y$ and $[\Lambda^{(t)}]$ as:

$$p_\theta(y, [\Lambda^{(t)}]) \triangleq \frac{1}{Z_\theta} \exp\left( \sum_{j=1}^{m} \sum_{i=1}^{n} (\theta_j \phi^{Acc}([\Lambda^{(t)}]_{ij}, y_i) + \theta_{j+m} \phi^{Lab}([\Lambda^{(t)}]_{ij}, y_i)) + \mathbf{b}_{y_i} \right) \tag{2}$$

We use Snorkel (Ratner et al., 2017) to learn $\theta$ by minimizing the negative log marginal likelihood given the observed $[\Lambda^{(t)}]$:

$$\hat{\theta}^{(t)} = \arg\min_{\theta} \sum_{i=1}^{n} -\log\left(\sum_{y\in\mathcal{Y}} p_\theta(y, [\Lambda^{(t)}]_i)\right) \tag{3}$$

We then set the accuracy weights of the LFs to be equal to $\hat{\theta}_j^{(t)}$.

**Update LF parameters $\tau$ keeping their accuracy weights fixed.** In this step, we use our label model (Eq. 1) to probabilistically label the training dataset, gather expert labels for the most uncertain data points, and finally update the LF parameters by minimizing the cross-entropy between the observed expert labels and probabilistic label predictions of the data points. In particular, we update the LF parameters $\tau$ via gradient back propagation, so as to minimize the negative log likelihood of all the expert labels observed so far:

$$\mathcal{L}_{CE} = \sum_{(\mathbf{x}_i, y_i)\in\mathbb{S}_l} \sum_{l=1}^{k} -\mathbb{1}_{\{l=y_i\}} \log \hat{\mathbf{p}}_{il} \tag{4}$$

where $\mathbb{S}_l$ is the set of expert labeled data points. While in our experiments, at each time step $t$, we gather an expert label for the data point with the highest entropy, our methods are generic and can seamlessly incorporate other active learning acquisition functions. Our algorithm is briefly summarised in Algorithm 1.

---

**Algorithm 1** WARM: The proposed active weak supervision algorithm.

1: **procedure** WARM($[\Gamma^{(1)}], \mathcal{S}_u = \{(\mathbf{x}_i, y_i)\}_{i=1}^{n}, \mathbf{b}$)
2:     **for** $t \in 1$ to $T$ **do**
3:         $[\Lambda^{(t)}] \leftarrow c([\Gamma^{(t)}])$                                          $\triangleright$ Convert to crisp votes
4:         $\hat{\theta}^{(t)} \leftarrow \arg\min_{\theta} -\log\left(\sum_{Y\in\mathcal{Y}} p_\theta(Y, [\Lambda^{(t)}])\right)$
5:         **for** $i \leftarrow 1$ to $|\mathbb{S}_u|$ **do**
6:             $\hat{\mathbf{p}}_{il} \leftarrow \mathbf{b}_l + \sum_{j=1}^{m} \gamma_j^{(t)}(\mathbf{x}_i, c)\hat{\theta}_j^{(t)}, \forall l \in \{1,\ldots,k\}$   $\triangleright$ Aggregate weighted LF votes
7:             $\hat{\mathbf{p}}_i \leftarrow \text{softmax}(\hat{\mathbf{p}}_i)$
8:             $\mathbf{E}_i \leftarrow -\sum_{l=1}^{k} \hat{\mathbf{p}}_{il} \log(\hat{\mathbf{p}}_{il})$                        $\triangleright$ Compute entropies
9:         **end for**
10:        $y_q \leftarrow \text{query\_user}(\{\mathbf{x}_q | q \leftarrow \arg\max_{i\in\mathbb{S}_u} \mathbf{E}\})$          $\triangleright$ Query user for label
11:        $\mathbb{S}_l \leftarrow \mathbb{S}_l \cup (\mathbf{x}_q, y_q); \mathbb{S}_u \leftarrow \mathbb{S}_u / (\mathbf{x}_q, y_q)$
12:        $\mathcal{L}_{CE} \leftarrow \sum_{(\mathbf{x}_i, y_i)\in\mathbb{S}_l} \sum_{l=1}^{k} -\mathbb{1}_{\{l=y_i\}} \log \hat{\mathbf{p}}_{il}$
13:        $\Gamma^{(t+1)} \leftarrow \text{backprop\_errors}(\Gamma^{(t)}, \mathcal{L}_{CE})$           $\triangleright$ Update LF parameters
14:     **end for**
15: **end procedure**

---

## 4 EXPERIMENTS

### 4.1 DATASETS AND WEAK SUPERVISION SOURCES

Like most prior work on active and automated weak supervision, we primarily focus on the binary classification setting for consistency (Varma & Ré, 2018; Biegel et al., 2021; Nashaat et al., 2018). Specifically, we carry out experiments on various real-world clinical classification datasets of varying complexity and sizes. See Table 1 for a summary of all the datasets used. All of the above datasets were taken from the UCI Machine Learning Repository (Dua & Graff, 2017). Refer Appendix A.1 for dataset details.

Since we did not have the requisite clinical expertise to define good weak supervision sources, we trained a Random Forest (RF) classifier of decision stumps (depth-one decision trees) to automatically to simulate expert heuristics for simple LFs. Specifically, we trained a Random Forest

| Dataset | Classification task | # Train | # Test | # Features | # LFs | # Classes | # +ve class (%) |
|---|---|---|---|---|---|---|---|
| *Wisconsin Breast Cancer* | Benign / malignant cancer | 559 | 140 | 9 | 10 | 2 | 34 |
| *Wisconsin Diagnostic Breast Cancer* | Benign / malignant cancer | 455 | 114 | 30 | 10 | 2 | 37 |
| *Heart Disease* | Absence / Presence of heart disease | 736 | 184 | 13 | 5 | 2 | 55 |
| *Diabetes* | Patient was re-admitted or not | 81412 | 20354 | 47 | 10 | 2 | 46 |
| *EEG* | Clinical findings in EEG | 4228 | 1058 | 26 | 17 | 4 | (37, 25, 27, 11) |
| *Synthetic* | Gaussian clusters | 10000 | 3000 | 2 | 3 | 2 | 50 |

Table 1: Summary of the six classification datasets used for evaluation.

classifier with 5 or 10 decision stumps on a $50\%$ sub-sample of the training set, such that each decision stump served as a LF. We used the Random Forest Classifier implementation from the `Scikit-Learn` (Pedregosa et al., 2011) Python library for the same. For all datasets, we acquired a set of 10 LFs, with the exception of the diabetes dataset for which we used a set of 5 LFs only. Appendix A.6 contains examples of LFs for the Heart Disease dataset.

As in Biegel et al. (2021), we also benchmark `WARM` on a *synthetic dataset* comprising of two classes modeled by 2-dimensional isotropic Gaussians, with a training and testing set of 10,000 and 3,000 data points, respectively. We use the same set of 3 LFs as Biegel et al. (2021) for our experiments.

Finally, to subject `WARM` to even more realistic settings, we used it to iteratively refine expert-defined heuristics to classify clinical findings in electroencephalogram (EEG) recordings. The dataset comprised of continuous 6-hour amplitude-integrated EEG (aEEG)[3] recordings from 1310 comatose patients resuscitated after suffering from cardiac arrest, and admitted to the Intensive Care Unit of a large tertiary care hospital between February 2010 and April 2019. The dataset was jointly annotated by two experts in cardiac arrest care and clinical EEG interpretation, using a simplified multinomial labeling convention to summarize EEG findings that occur commonly post cardiac arrest. Specifically, each raw EEG record was broken into six disjoint 1-hour windows and classified as generalized suppression; burst suppression with epileptiform activity (including burst suppression with identical bursts); burst suppression without epileptiform activity; and, near-continuous or continuous background activity (Elmer et al., 2016; 2020). We also developed a total of 17 LFs tailored to the prediction problem with help from one of the expert clinicians.

## 4.2 COMPARISON OF METHODS

We compared `WARM` with the two existing active weak supervision approaches, namely `Active WeaSuL` (Biegel et al., 2021) and the method proposed by Nashaat et al. (2018). All the methods had access to the same set of initial LFs and were run for 30 active learning iterations, where they were allowed to query one ground truth label from the training set at each iteration. We also benchmarked `WARM` against `Snuba` (Varma & Ré, 2018), a popular weak supervision approach which automatically synthesizes weak supervision sources from a small subset of labeled examples incorporates them with the data programming label model to probabilistically label data. We ran `Snuba` with a random labeled subset of 10, 20 and 30 data points with decision tree stumps as heuristics. To report test set performance, we trained downstream logistic regression models on the crisp versions of the probabilistic labels assigned by each method. We also compared these methods against a logistic regression (LR) model trained using a vanilla `active learning` sampling strategy based on the distance from the decision boundary as in Biegel et al. (2021), and a `fully-supervised` one trained using the entire labeled training set. The fully supervised LR model serves as a proxy for the maximum achievable performance with the given features, train-test split and downstream classifier.

Since existing online implementations of `Snuba`, `Active WeaSuL` and the method proposed by Nashaat et al. (2018) were designed exclusively for binary classification problems, we were only able to compare `WARM` with active learning and fully supervised baselines on the EEG dataset involving multi-class classification. For all our experiments, we used the logistic regression model implementation from `Scikit-Learn` (Pedregosa et al., 2011), trained for a maximum of 500 iterations and optimized using LBFGS. `WARM` models were trained and built using PyTorch 1.8.1 Paszke et al. (2019) along with Python 3.8.1. All our experiments were carried out on a Linux desktop with a typical machine having 8 Intel(R) Core(TM) i7-6700K CPUs and 16 GB of RAM.

---

[3]aEEG is a widely used quantitative summary of multi-channel EEG data.

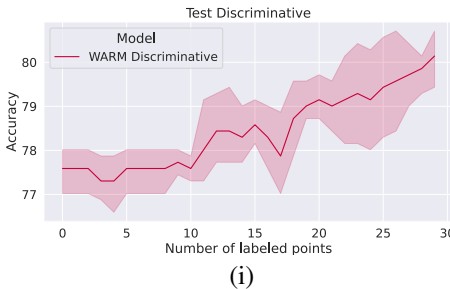 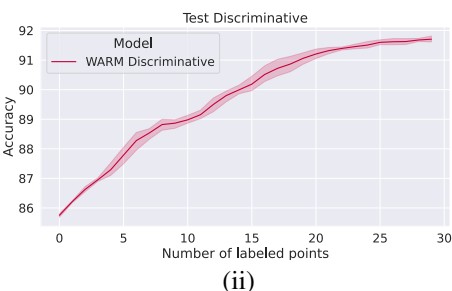

(i)                                      (ii)

Figure 2: Knowledge shift experiments on the (i) *heart disease* and (ii) *synthetic* datasets. The accuracy of end models increases with more active learning iterations as `WARM` adapts LFs and the resulting label model to the target population.

## 4.3 EXPERIMENTAL DESIGN

We aim to answer the following research questions through our experiments.

*Does* `WARM` *improve the quality of the training dataset and how does this improvement translate to downstream classifier performance?* We evaluate the quality of the training dataset by comparing the crisp versions of the probabilistic labels assigned by all the weakly supervised approaches with the ground truth training labels.

*Can* `WARM` *combat the problem of knowledge shift and overcome noise in the initial LFs?* To this end, we carry out experiments on the *synthetic* and *Heart Disease* dataset. To simulate knowledge shift in the *synthetic dataset*, we start with the three LFs tailored to the original (source) data distribution, and introduce two other (target) gaussian clusters with different centroids. For the *heart disease* dataset, we automatically acquire LFs from the young population *i.e.* all patients with less than median age. We then train and evaluate `WARM` on disjoint subsets of old patient population. Since weakly supervised models rely heavily on the quality of its weak supervision sources, we conduct additional experiments on the *Heart Disease* dataset by adding varying levels of uniform random noise, scaled by the range of the features, to the LF decision parameters initially obtained using the random forest. I.e. 100% noise added could mean that the decision stump threshold is off by 100%.

## 5 RESULTS AND DISCUSSION

Table 2 summarizes the results of our experiments. The **Train** results represent the performance of the weak label models in comparison to ground truth training labels, and hence are a direct measure of label model quality. The **Test** results on the other hand, are from LR models trained on the crisp equivalents of the probabilistic labels and evaluated on the held-out test set. The latter set of results primarily reflect the quality of the downstream models trained using weak labels. We refer the

| | Methods | Synthetic | Heart Disease | Diagnostic BC | Wisconsin BC | Diabetes | EEG |
|---|---|---|---|---|---|---|---|
| **Train** | `Active WeaSuL` | $91.05 \pm 0.00$ | $47.01 \pm 0.54$ | $67.69 \pm 0.00$ | $67.69 \pm 0.00$ | $53.94 \pm 0.14$ | – |
| | **Nashaat et al. (2018)** | $90.70 \pm 0.01$ | $48.23 \pm 1.17$ | $65.36 \pm 0.11$ | $91.63 \pm 0.21$ | $53.98 \pm 0.21$ | – |
| | `Snuba` | $\mathbf{95.83 \pm 0.39}$ | $75.62 \pm 2.34$ | $92.53 \pm 0.90$ | $\mathbf{94.17 \pm 1.69}$ | $52.06 \pm 4.06$ | – |
| | `WARM` | $95.39 \pm 0.22$ | $\mathbf{78.78 \pm 1.35}$ | $\mathbf{93.05 \pm 0.90}$ | $94.10 \pm 0.00$ | $\mathbf{60.19 \pm 0.24}$ | $71.16 \pm 2.09$ |
| **Test** | `Active WeaSuL` | $95.51 \pm 0.52$ | $50.43 \pm 1.55$ | $88.95 \pm 1.53$ | $93.57 \pm 1.28$ | $54.57 \pm 0.41$ | – |
| | **Nashaat et al. (2018)** | $92.93 \pm 0.28$ | $53.59 \pm 7.07$ | $94.74 \pm 1.57$ | $92.29 \pm 1.05$ | $54.27 \pm 0.32$ | – |
| | `Snuba` | $95.51 \pm 0.37$ | $71.30 \pm 2.02$ | $88.95 \pm 0.43$ | $92.86 \pm 1.63$ | $52.14 \pm 4.28$ | – |
| | `WARM` | $95.77 \pm 0.20$ | $\mathbf{77.72 \pm 0.34}$ | $91.23 \pm 0.96$ | $89.29 \pm 0.00$ | $\mathbf{60.14 \pm 0.20}$ | $\mathbf{72.29 \pm 1.48}$ |
| | `Active Learning` | $\mathbf{97.33 \pm 0.15}$ | $76.63 \pm 1.33$ | $\mathbf{97.19 \pm 0.35}$ | $\mathbf{94.86 \pm 0.70}$ | $52.21 \pm 1.41$ | $64.50 \pm 1.80$ |
| | `Fully Sup.  LR` | $97.50$ | $77.72$ | $99.12$ | $95.71$ | $62.38$ | $73.25$ |

Table 2: Final accuracy of models on all the datasets together with 95% confidence intervals computed over 5 bootstrapping iterations. `WARM` improves the quality of training data as measured by the accuracy of label model on the **Train** set. This improvement further translates into better downstream model performance on the **Test** set. `Fully Sup.  LR` represents the performance of the fully-supervised logistic regression (LR) model trained with full access to ground truth labels.

interested reader to Appendix A.7 for supplementary results with additional metrics such as F1 and percentage change in accuracy. Our primary findings from the experiments are summarized below.

**`WARM` can improve the quality of training data.** We found that on all the datasets, the probabilistic labels assigned post active weak supervision via `WARM` are significantly more accurate than the ones assigned by compared methods. However, we did observe some variance in accuracy and its improvement across different datasets, primarily due to the nature of the data and the quality of initial weak supervision sources. For instance on the breast cancer data, `WARM` improved the accuracy of the label model by over $7\%$ across 30 active learning iterations. On the other hand, `WARM` could not improve the train generative performance on the heart disease data, but it also starts from a better higher accuracy. We observed similar trends for downstream model performance, albeit with smaller accuracy improvements. The smaller accuracy improvements can be attributed to the generalization capabilities of the downstream model over the labels it is trained on. Nevertheless, our findings underscore the importance of high quality labeled training sets. On the EEG dataset (Table 2 and Fig. 6–7), `WARM` achieved a $4 - 5\%$ increase in both label model and downstream classifier performance. These results are significant since even the initial set of LFs were developed by expert clinicians.

Moreover, we also found that traditional `Active Learning` strategy outperformed `WARM` on simpler classification tasks such as breast cancer diagnosis. However, for more complex prediction problems such as the predicting readmission on the *Diabetes dataset* and classifying clinical findings on the *EEG dataset*, active weak supervision via `WARM` significantly outperformed end models trained using active learning. Finally, experiment runtimes suggest that `WARM` is significantly faster than existing active weak supervision techniques across datasets (Appendix A.5).

**`WARM` can help combat knowledge shift and de-noise LFs.** Results from knowledge shift experiments on the *heart disease* and *diabetes dataset* in Fig. 2 and Table 4,5 reveal that `WARM` can help both weak labelers and the downstream classifier adapt to different population characteristics. We observed approximately a $1 - 3\%$ and $4 - 6\%$ increase in label and end model performance on the *heart disease* and *synthetic datasets*, respectively. See Appendix A.6 for all LFs defined on the *heart disease dataset* to detect the presence of heart disease in the young and old population.

Fig. 3(ii) and Tab. 6 summarizes the performance of `WARM` across 50 active learning iterations, when varying levels of random noise is artificially added to LFs defined on the *Wisconsin Diagnostic Breast Cancer dataset*. Our results clearly show that noisy LFs hurt both label and end model performance. However, `WARM` can iteratively de-noise LFs by actively collecting a few more labeled examples. Even with $75\%$ noise initially, `WARM` can refine LFs such that the resulting label and end model perform on par with those trained using LFs with no noise. Additionally, `WARM` consistently outperforms the other active weak supervision baselines[4].

**Changes to LFs brought about by `WARM` are interpretable.** If the LFs themselves are inherently interpretable, `WARM` supports their interpretability by enabling domain experts to track changes in their decision parameters. Fig. 3(ii) for instance shows LFs before and after refinement by `WARM` on the *synthetic dataset*. We also draw the reader's attention to the knowledge shift experiments on the *heart disease dataset* for more examples of interpretable changes to LF parameters. Appendix A.6 lists all the LFs tailored to the young and old population. For example, the rule `If exercise induced angina <= 144.666(150.123) then ABSENCE else PRESENCE` clearly says that while exercise induced angina higher than 144.67 was indicative of heart disease in the young population, older people tend to naturally have higher values resulting in a higher decision threshold.

**Ablation experiments on `WARM` shows that it benefits from both active learning and weak supervision**. We also perform ablation experiments to investigate if `WARM` is performing as expected. In Appendix A.8, we see that `WARM` directly benefits from Uncertainty Sampling from its active learning component, outperforming the random sampling in every case. From Appendix A.10,A.9 it is generally able to generally outperform `Active Learning` on complex or large datasets due to its Weak Supervision component.

---

[4]Note that active learning results are not affected because we only add noise to LFs, which are only used by weak supervision methods

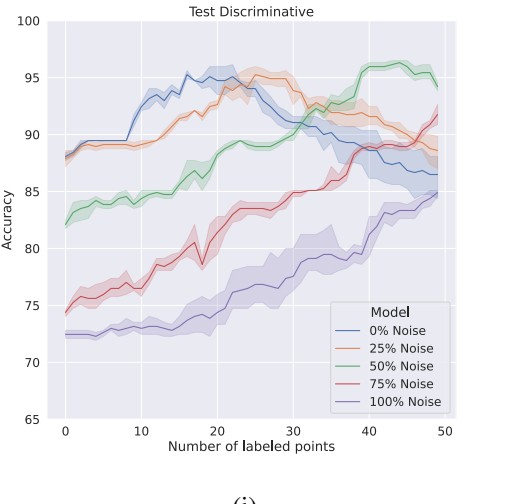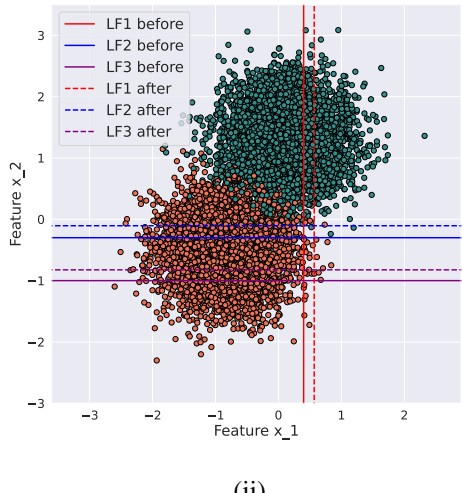

Figure 3: (i): End model testing accuracy on the *Wisconsin Diagnostic Breast Cancer dataset* as a function of uniform random noise and active learning iterations. With a few expert-labeled data points WARM can effectively de-noise LFs to eventually improve end model performance. (ii): Plot of the *synthetic dataset* and its three LFs before and after 30 active refinement iterations. WARM supports interpretability by allowing experts to track changes in LFs.

## 6 CONCLUSION

With more medical data being collected, methods that address the lack of labeled data in clinical applications of machine learning are becoming increasingly vital. If the vast amounts of unlabeled clinical data could be utilized to its full potential, the possibility of augmenting physicians with ML to increase positive patient outcomes is substantial. In addition, a lot of of useful information can be provided to ML systems from expert guidance, whether it be via heuristic design or via hand-labeling individual data points; making such capabilities efficient and effective is greatly needed in practice. WARM (Active Refinement of Weakly Supervised Models) is a fast active learning addition to data programming that substantially and inexpensively improves the empirical accuracy of training labels and downstream classifiers. Our experiments show that WARM has the capability to address the common problems of knowledge shift and rule denoising with minimal additional labeled data. If ran on interpretable LFs, WARM is even able to reveal strategic, data-driven, interpretable updates to the LF thresholds and parameters. Limitations of WARM include its reliance on differentiable and reasonably accurate initial LFs, which may occasionally be more expensive to acquire than pointillistic labels. However, we have shown that most frequently used classes of LFs are intrinsically differentiable or can be made so. WARM is already a step towards more efficient and effective algorithms to tackle labeled-data-scarce ML problems in healthcare and beyond.

Additional experiments would benefit precise assessment of properties and scoping of practical utility of WARM in a wider range of its possible applications. Future work should also involve user studies to gauge attainable efficiency gains and utility of interpretable adjustments to the LFs, and whether they meaningfully boost perceived usability of the weak supervision process in practice. Other future directions could include using meta-AL to learn query selection strategies based off of previous AL outcomes as in (Konyushkova et al., 2017; Bachman et al., 2017) or AL with partial feedback approaches as in (Hu et al., 2018). Another challenge to address is mitigating overfitting to the active learning samples; we could perform early stopping, learning rate decay, gradient clipping, or cross validation to better generalize our models. Furthermore, we would like to research the connection between our framework and automated (Varma & Ré, 2018) or active (Boecking et al., 2020) discovery of new LFs.

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

# A    APPENDIX

## A.1    DATASET DETAILS

The *Heart Disease dataset* (Detrano et al., 1989) contains clinical and noninvasive test results from patients undergoing angiography at the Cleveland Clinic in Cleveland, Ohio, Hungarian Institute of Cardiology in Budapest, Hungary, Veterans Administration Medical Center in Long Beach, California; and University Hospitals in Zurich and Basel, Switzerland. For our experiments, we concatenate all the databases into one and predict whether or not an individual has heart disease. The *Diabetes dataset* (Strack et al., 2014) comprises of clinical records from over million patients admitted to 130 US hospitals and integrated delivery networks between 1999 to 2008. Given features representing patients such as their age, race, gender, time in hospital, diagnosis etc., the goal is to predict whether they are re-admitted for further treatment. We also use two breast cancer datasets, namely *Wisconsin Diagnostic Breast Cancer dataset* (Street et al., 1993) and *Wisconsin Breast Cancer Database* (Mangasarian & Wolberg, 1990). For both the datasets, the prediction task is to distinguish between samples of malignant and benign breast cancer using different sets of features. The diagnostic dataset comprises of features characterizing cell nuclei present in a digitized image of a fine needle aspirate (FNA) of a breast mass. The latter dataset instead contains 9 cytological characteristics of breast FNA graded between 1–10 at the time of sample collection.

## A.2    EEG DATA COLLECTION

EEG monitoring was initiated a median of 10 hours after initial collapse as standard routine of care on arrival to the intensive care unit (Elmer et al., 2016). EEG was recorded using XLTech Natus Neuroworks digital video/EEG systems (Natus Medical Inc) at 256Hz from 22 gold-plated cup electrodes placed in the international 10-20 system positions.

Löfhede et al. (2008) has shown near-perfect spatial correlation of many clinically important EEG features in patients after cardiac arrest, allowing the 22-sample signal to be down-sampled without significant information loss. Additionally, after cardiac arrest, many prognostic clinical findings can

be identified accurately from lower-resolution quantitative summaries (e.g. 1Hz aEEG, Fig. 4) (Oh et al., 2015; Glass et al., 2013). Thus, to optimize computational efficiency, we aimed to develop similar heuristics as labeling functions to classify raw EEG signals from contemporaneous aEEG summaries.

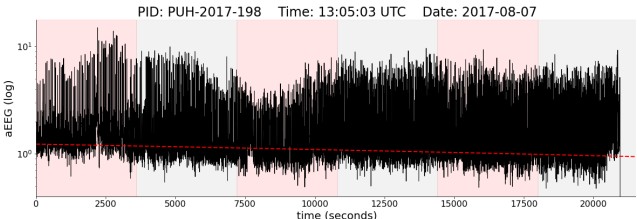

Figure 4: An example aEEG waveform ($T = 21600$ seconds). Each alternating pink and grey region marks an one-hour segment. The dashed red line represents baseline aEEG value determined using robust linear regression (aka RANSAC (Fischler & Bolles, 1981)).

| Characteristic | Number (%) |
|---|---|
| Number of patients | 1310 |
| Age, mean (std), years | 57 (17) |
| Female | 503 (38) |
| Initial arrest rhythm | |
|     VT/VF | 370 (28) |
|     PEA | 482 (37) |
|     Asystole | 371 (28) |
|     Unknown | 87 (6) |
| Arrest location | |
|     In-hospital cardiac arrest | 243 (19) |
|     Out-of-hospital cardiac arrest | 1067 (81) |
| Hours from arrest to EEG start, median (IQR) | 10 (6) |
| Survived to hospital discharge | 387 (30) |
| Disposition | |
|     Home | 111 (29) |
|     Acute rehabilitation | 112 (29) |
|     Skilled nursing facility | 78 (20) |
|     Long-term acute care | 46 (12) |
|     Hospice | 22 (6) |
|     Other | 18 (4) |
| Circumstances of death | |
|     Rearrested, intractable shock, multi-system organ failure | 184 (20) |
|     Withdrawal for non-neurological reasons | 114 (10) |
|     Brain death | 96 (12) |
|     Withdrawal for perceived poor neurological prognosis | 529 (57) |

Table 3: Clinical Characteristics and Patient Outcomes: A summary of the patients characteristics in the *EEG dataset*.

For our experiments, we used 6-hour aEEG time series starting at the initiation of EEG monitoring from 1310 patients with mean age 57 years of whom 503 (38%) were female. Approximately 30% survived to discharge from the hospital. A large subset of our data (approximately 70%) was initially jointly annotated by two experts in cardiac arrest care and clinical EEG interpretation at UPMC. They manually annotated EEG findings using an adaptation of the 2016 American Clinical Neurophysiology Society guidelines (Hirsch et al., 2013).

Based on prior research and our expert clinicians, we developed a simplified multinomial labeling convention to summarize prognostic EEG findings that occur commonly after cardiac arrest. Specifically, we classified the raw EEG waveforms as: generalized suppression; burst suppression with epileptiform activity (including burst suppression with identical bursts); burst suppression without epileptiform activity; and, near-continuous or continuous background activity (Elmer et al., 2020). Because continuous background activity with epileptiform activity occurred infrequently in this data set, we did not further subdivide categories of patients with continuous background activity.

## A.3    KNOWLEDGE SHIFT EXPERIMENTS

| Synthetic Dataset | | | | |
|---|---|---|---|---|
| | Methods | Accuracy | ΔAccuracy | F1 |
| **Train Generative** | Active WeaSuL | $90.29 \pm 7.57$ | $17.77 \pm 7.57$ | $88.94 \pm 9.30$ |
| | Nashaat et al. | $71.17 \pm 3.20$ | $-1.35 \pm 3.20$ | $68.80 \pm 4.78$ |
| | WARM | $91.66 \pm 0.05$ | $4.72 \pm 0.07$ | $91.85 \pm 0.03$ |
| **Test Discriminative** | Active WeaSuL | $92.83 \pm 2.80$ | $9.47 \pm 2.80$ | $92.46 \pm 3.03$ |
| | Nashaat et al. | $82.42 \pm 3.22$ | $-0.95 \pm 3.22$ | $80.62 \pm 4.94$ |
| | Active learning | $96.85 \pm 0.09$ | $46.85 \pm 0.09$ | $96.85 \pm 0.08$ |
| | WARM | $91.71 \pm 0.11$ | $5.95 \pm 0.15$ | $91.60 \pm 0.14$ |

Table 4: Knowledge Shift experiments on the *Synthetic* Dataset. Accuracy is final accuracy, and ΔAccuracy is change from initial to final accuracy.

| Heart Disease Dataset | | | | |
|---|---|---|---|---|
| | Models | Accuracy | ΔAccuracy | F1 |
| **Train Generative** | Active WeaSuL | $36.31 \pm 0.00$ | $0.00 \pm 0.00$ | $5.47 \pm 0.38$ |
| | Nashaat et al. | $37.26 \pm 0.52$ | $0.95 \pm 0.52$ | $8.02 \pm 1.05$ |
| | WARM | $80.56 \pm 1.88$ | $1.06 \pm 0.65$ | $86.53 \pm 1.27$ |
| **Test Discriminative** | Active WeaSuL | $30.35 \pm 2.07$ | $4.82 \pm 2.07$ | $13.16 \pm 8.18$ |
| | Nashaat et al. | $35.32 \pm 3.15$ | $9.79 \pm 3.15$ | $21.27 \pm 7.59$ |
| | Active learning | $79.01 \pm 1.65$ | $29.01 \pm 1.65$ | $85.65 \pm 1.42$ |
| | WARM | $80.14 \pm 0.78$ | $2.55 \pm 1.24$ | $86.97 \pm 0.33$ |

Table 5: Knowledge Shift experiments on the *Heart Disease* Dataset. Accuracy is final accuracy, and ΔAccuracy is change from initial to final accuracy.

## A.4   ADDING ARTIFICIAL NOISE TO LFs

| Heart Disease Dataset | | | |
|---|---|---|---|
| Methods | Accuracy | $\Delta$Accuracy | F1 |
| **0% Noise** | | | |
| **Train Generative** | | | |
| Active WeaSuL | $46.20 \pm 0.54$ | $0.27 \pm 0.54$ | $4.90 \pm 1.14$ |
| Nashaat et al. | $48.23 \pm 1.17$ | $2.31 \pm 1.17$ | $11.75 \pm 4.06$ |
| WARM | $78.61 \pm 1.56$ | $0.00 \pm 0.00$ | $80.50 \pm 1.46$ |
| **Test Discriminative** | | | |
| Active WeaSuL | $48.26 \pm 1.59$ | $5.87 \pm 1.59$ | $28.25 \pm 3.45$ |
| Nashaat et al. | $53.59 \pm 7.07$ | $11.20 \pm 7.07$ | $38.54 \pm 14.93$ |
| WARM | $78.59 \pm 0.43$ | $0.00 \pm 0.00$ | $82.64 \pm 0.45$ |
| **25% Noise** | | | |
| **Train Generative** | | | |
| Active WeaSuL | $48.34 \pm 0.53$ | $0.65 \pm 0.53$ | $12.00 \pm 2.12$ |
| Nashaat et al. | $50.11 \pm 0.83$ | $2.42 \pm 0.83$ | $19.22 \pm 3.79$ |
| WARM | $79.62 \pm 1.09$ | $-0.05 \pm 0.35$ | $81.39 \pm 1.25$ |
| **Test Discriminative** | | | |
| Active WeaSuL | $58.26 \pm 1.47$ | $10.43 \pm 1.47$ | $49.03 \pm 2.56$ |
| Nashaat et al. | $59.13 \pm 5.34$ | $11.30 \pm 5.34$ | $50.36 \pm 9.79$ |
| WARM | $77.83 \pm 0.72$ | $0.54 \pm 0.60$ | $81.84 \pm 0.87$ |
| **50% Noise** | | | |
| **Train Generative** | | | |
| Active WeaSuL | $47.15 \pm 0.00$ | $0.00 \pm 0.00$ | $10.16 \pm 0.00$ |
| Nashaat et al. | $48.26 \pm 0.25$ | $1.11 \pm 0.25$ | $15.00 \pm 0.74$ |
| WARM | $78.89 \pm 0.96$ | $-0.33 \pm 0.83$ | $80.78 \pm 0.94$ |
| **Test Discriminative** | | | |
| Active WeaSuL | $46.09 \pm 1.11$ | $5.33 \pm 1.11$ | $21.67 \pm 3.55$ |
| Nashaat et al. | $52.07 \pm 7.89$ | $11.30 \pm 7.89$ | $31.31 \pm 18.65$ |
| WARM | $77.17 \pm 1.50$ | $-1.41 \pm 1.12$ | $80.84 \pm 1.58$ |
| **75% Noise** | | | |
| **Train Generative** | | | |
| Active WeaSuL | $43.29 \pm 1.47$ | $-0.73 \pm 1.47$ | $15.23 \pm 2.84$ |
| Nashaat et al. | $48.45 \pm 4.21$ | $4.43 \pm 4.21$ | $25.20 \pm 6.44$ |
| WARM | $79.54 \pm 1.25$ | $0.43 \pm 0.95$ | $81.79 \pm 1.16$ |
| **Test Discriminative** | | | |
| Active WeaSuL | $42.61 \pm 0.81$ | $1.30 \pm 0.81$ | $11.35 \pm 2.56$ |
| Nashaat et al. | $48.80 \pm 7.89$ | $7.50 \pm 7.89$ | $25.01 \pm 21.41$ |
| WARM | $77.61 \pm 0.41$ | $1.30 \pm 0.55$ | $81.53 \pm 0.58$ |
| **100% Noise** | | | |
| **Train Generative** | | | |
| Active WeaSuL | $49.86 \pm 0.00$ | $0.00 \pm 0.00$ | $35.83 \pm 0.00$ |
| Nashaat et al. | $53.26 \pm 2.31$ | $3.40 \pm 2.31$ | $35.62 \pm 4.92$ |
| WARM | $78.42 \pm 1.21$ | $0.82 \pm 0.63$ | $79.00 \pm 0.77$ |
| **Test Discriminative** | | | |
| Active WeaSuL | $37.83 \pm 2.24$ | $-1.30 \pm 2.24$ | $13.52 \pm 4.19$ |
| Nashaat et al. | $54.35 \pm 5.00$ | $15.22 \pm 5.00$ | $42.74 \pm 9.75$ |
| WARM | $71.96 \pm 1.31$ | $-1.41 \pm 0.55$ | $74.64 \pm 1.36$ |

Table 6: Experiments on the *Heart Disease dataset* by adding varying levels of uniform random noise to LF decision parameters. Accuracy is final accuracy, and $\Delta$Accuracy is change from initial to final accuracy. Noisy LFs hurt both label and end model performance. However, WARM can iteratively de-noise LFs by actively collecting a few more labeled examples. Even with 75% noise initially, WARM can refine LFs such that the resulting label and end model perform on par with those trained using LFs with no noise. WARM also consistently outperforms other baselines.

## A.5 RUNTIMES

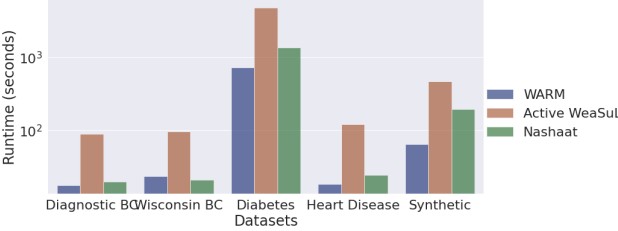

Figure 5: Active weak supervision runtimes (in seconds on log scale). WARM is significantly faster than compared existing methods across all datasets.

A.6 EXAMPLE RULES

On the *heart disease dataset*, we carried out our knowledge shift experiments, where the initial labeling functions were based on the young population, but the target population was that of old patients. Below is a list of initial rules tailored to the young patients. The decision parameters in parenthesis reflect the modified labeling function parameters, tailored to older patients.

```
If Chest pain type <= 0.898(0.978) then ABSENCE else PRESENCE
If Serum cholesterol in mg/dl <= 124.057(119.793) then PRESENCE else ABSENCE
If Maximum heart rate achieved <= 0.244(0.079) then PRESENCE else ABSENCE
If Exercise induced angina <= 144.666(150.123) then ABSENCE else PRESENCE
If thal <= 0.646(0.796) then ABSENCE else PRESENCE
If Chest pain type <= 0.898(0.978) then ABSENCE else PRESENCE
If ST depression induced by exercise relative to rest <= 0.273(0.383)
    then ABSENCE else PRESENCE
If Exercise induced angina <= 144.666(150.123) then ABSENCE else PRESENCE
If thal <= 0.646(0.796) then ABSENCE else PRESENCE
If Maximum heart rate achieved <= 0.533(0.367) then PRESENCE else ABSENCE
```

## A.7 WARM COMPARISON TO OTHER MODELS

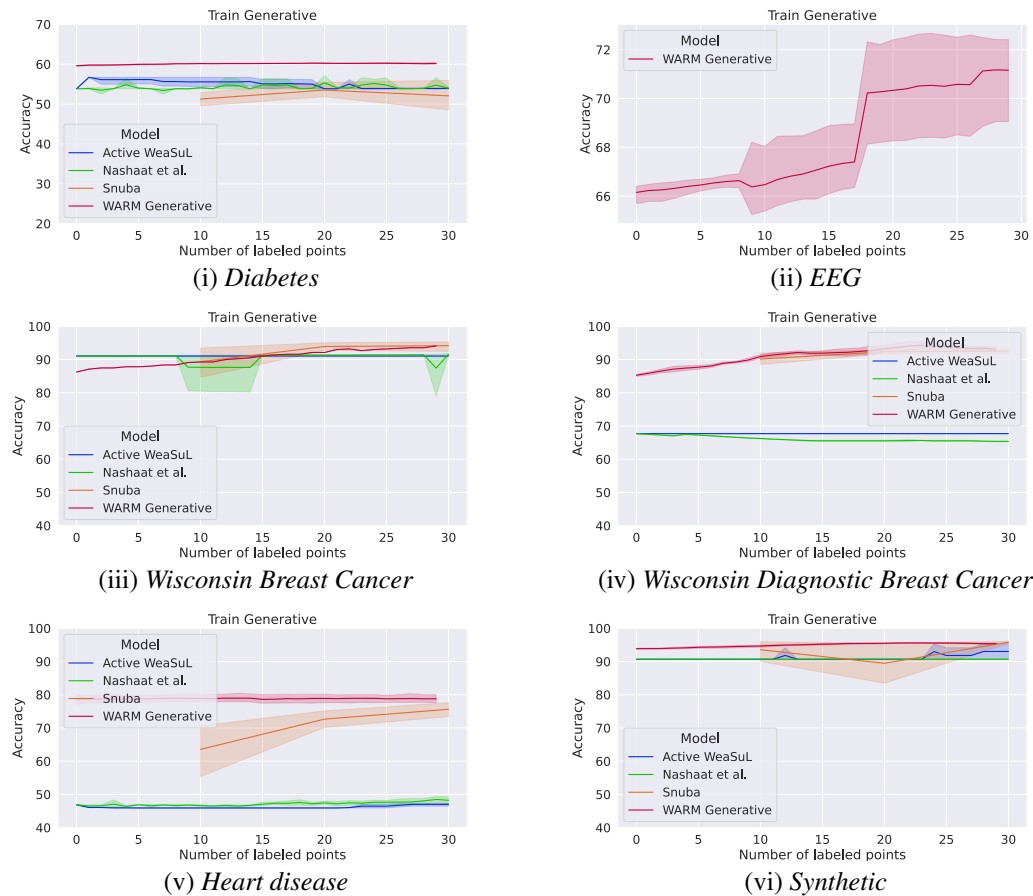

(i) *Diabetes*

(ii) *EEG*

(iii) *Wisconsin Breast Cancer*

(iv) *Wisconsin Diagnostic Breast Cancer*

(v) *Heart disease*

(vi) *Synthetic*

Figure 6: Results of label model predictions on the training datasets. These results are a direct measure of training set quality.

| | | Wisconsin Breast Cancer Database | | |
|---|---|---|---|---|
| | **Methods** | Accuracy | $\Delta$Accuracy | F1 |
| **Train Generative** | Active WeaSuL (Biegel et al., 2021) | $67.69 \pm 0.00$ | $0.00 \pm 0.00$ | $85.88 \pm 0.00$ |
| | Nashaat et al. (2018) | $91.63 \pm 0.21$ | $0.57 \pm 0.21$ | $86.75 \pm 0.40$ |
| | Snuba (Varma & Ré, 2018) | $\mathbf{94.17 \pm 1.69}$ | $4.83 \pm 4.24$ | $\mathbf{91.40 \pm 2.71}$ |
| | WARM (Ours) | $94.10 \pm 0.00$ | $\mathbf{7.87 \pm 0.00}$ | $91.08 \pm 0.02$ |
| **Test Discriminative** | Active WeaSuL (Biegel et al., 2021) | $93.57 \pm 1.28$ | $0.00 \pm 1.28$ | $90.20 \pm 1.95$ |
| | Nashaat et al. (2018) | $92.29 \pm 1.05$ | $-1.29 \pm 1.05$ | $88.08 \pm 1.79$ |
| | Snuba (Varma & Ré, 2018) | $92.86 \pm 1.63$ | $3.29 \pm 3.77$ | $88.71 \pm 2.98$ |
| | WARM (Ours) | $89.29 \pm 0.00$ | $1.07 \pm 0.36$ | $82.76 \pm 0.00$ |
| | Active Learning | $94.86 \pm 0.70$ | $44.86 \pm 0.70$ | $92.39 \pm 1.11$ |
| | FS Logistic Regression | | $-$ | |

Table 7: Performance metrics for the *Wisconsin Breast Cancer database*. Accuracy is final accuracy, and $\Delta$Accuracy is change from initial to final accuracy.

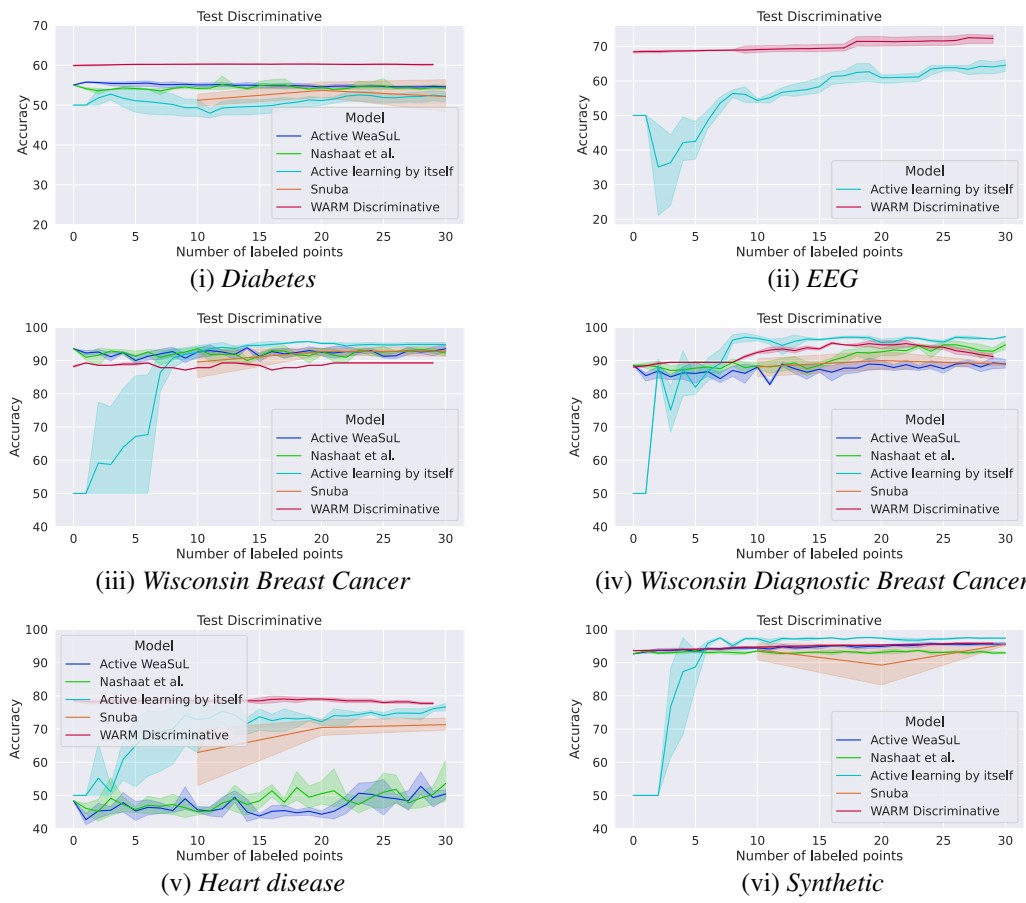

Figure 7: Results of the logistic regression end model evaluated on the testing dataset. datasets.

| | **Wisconsin Diagnostic Breast Cancer Dataset** | | |
| | **Methods** | Accuracy | ΔAccuracy | F1 |
|---|---|---|---|---|
| **Train Generative** | Active WeaSuL (Biegel et al., 2021) | $67.69 \pm 0.00$ | $0.00 \pm 0.00$ | $28.29 \pm 0.00$ |
| | Nashaat et al. (2018) | $65.36 \pm 0.11$ | $-2.33 \pm 0.11$ | $18.93 \pm 0.46$ |
| | Snuba (Varma & Ré, 2018) | $92.53 \pm 0.90$ | $2.37 \pm 1.69$ | $90.07 \pm 0.86$ |
| | WARM (Ours) | $\mathbf{93.05 \pm 0.90}$ | $\mathbf{7.78 \pm 1.25}$ | $\mathbf{90.52 \pm 1.74}$ |
| **Test Discriminative** | Active WeaSuL (Biegel et al., 2021) | $88.95 \pm 1.53$ | $0.35 \pm 1.53$ | $79.28 \pm 3.23$ |
| | Nashaat et al. (2018) | $94.74 \pm 1.57$ | $6.14 \pm 1.57$ | $90.83 \pm 3.01$ |
| | Snuba (Varma & Ré, 2018) | $88.95 \pm 0.43$ | $0.88 \pm 3.09$ | $82.15 \pm 0.39$ |
| | WARM (Ours) | $91.23 \pm 0.96$ | $3.16 \pm 1.31$ | $87.27 \pm 1.19$ |
| | Active Learning | $97.19 \pm 0.35$ | $47.19 \pm 0.35$ | $95.45 \pm 0.59$ |
| | FS Logistic Regression | | − | |

Table 8: Performance metrics for the *Wisconsin Diagnostic Breast Cancer* dataset. Accuracy is final accuracy, and ΔAccuracy is change from initial to final accuracy.

| Heart Disease Dataset | | | | |
|---|---|---|---|---|
| | **Methods** | Accuracy | ΔAccuracy | F1 |
| **Train Generative** | Active WeaSuL (Biegel et al., 2021) | $47.01 \pm 0.54$ | $0.14 \pm 0.54$ | $6.61 \pm 1.14$ |
| | Nashaat et al. (2018) | $48.23 \pm 1.17$ | $1.36 \pm 1.17$ | $11.75 \pm 4.06$ |
| | Snuba (Varma & Ré, 2018) | $75.62 \pm 2.34$ | $\mathbf{12.07 \pm 9.97}$ | $75.95 \pm 4.48$ |
| | WARM (Ours) | $\mathbf{78.78 \pm 1.35}$ | $0.08 \pm 0.24$ | $\mathbf{80.52 \pm 1.24}$ |
| **Test Discriminative** | Active WeaSuL (Biegel et al., 2021) | $50.43 \pm 1.55$ | $2.07 \pm 1.55$ | $33.21 \pm 4.14$ |
| | Nashaat et al. (2018) | $53.59 \pm 7.07$ | $5.22 \pm 7.07$ | $38.54 \pm 14.93$ |
| | Snuba (Varma & Ré, 2018) | $71.30 \pm 2.02$ | $8.37 \pm 9.56$ | $73.01 \pm 4.73$ |
| | WARM (Ours) | $77.72 \pm 0.34$ | $-0.87 \pm 0.27$ | $81.74 \pm 0.55$ |
| | Active Learning | $76.63 \pm 1.33$ | $26.63 \pm 1.33$ | $79.34 \pm 1.17$ |
| | FS Logistic Regression | | $-$ | |

Table 9: Performance metrics for the *Heart disease dataset*. Accuracy is final accuracy, and ΔAccuracy is change from initial to final accuracy.

| Diabetes Dataset | | | | |
|---|---|---|---|---|
| | **Methods** | Accuracy | ΔAccuracy | F1 |
| **Train Generative** | Active WeaSuL (Biegel et al., 2021) | $53.94 \pm 0.14$ | $0.10 \pm 0.14$ | $7.77 \pm 7.32$ |
| | Nashaat et al. (2018) | $53.98 \pm 0.21$ | $0.14 \pm 0.21$ | $7.35 \pm 4.55$ |
| | Snuba (Varma & Ré, 2018) | $52.06 \pm 4.06$ | $0.81 \pm 5.43$ | $50.16 \pm 5.29$ |
| | WARM (Ours) | $60.19 \pm 0.24$ | $0.59 \pm 0.29$ | $48.53 \pm 2.16$ |
| **Test Discriminative** | Active WeaSuL (Biegel et al., 2021) | $54.57 \pm 0.41$ | $-0.47 \pm 0.41$ | $4.71 \pm 3.73$ |
| | Nashaat et al. (2018) | $54.27 \pm 0.32$ | $-0.76 \pm 0.32$ | $2.62 \pm 2.37$ |
| | Snuba (Varma & Ré, 2018) | $52.14 \pm 4.28$ | $0.94 \pm 5.48$ | $50.04 \pm 5.94$ |
| | WARM (Ours) | $60.14 \pm 0.20$ | $0.21 \pm 0.31$ | $48.15 \pm 2.44$ |
| | Active Learning | $52.21 \pm 1.41$ | $2.21 \pm 1.41$ | $50.23 \pm 7.44$ |
| | FS Logistic Regression | | $-$ | |

Table 10: Performance metrics for the *Diabetes dataset*. Accuracy is final accuracy, and ΔAccuracy is change from initial to final accuracy.

| Synthetic Dataset | | | | |
|---|---|---|---|---|
| | **Methods** | Accuracy | ΔAccuracy | F1 |
| **Train Generative** | Active WeaSuL (Biegel et al., 2021) | $91.05 \pm 0.00$ | $0.00 \pm 0.00$ | $85.88 \pm 0.00$ |
| | Nashaat et al. (2018) | $90.70 \pm 0.01$ | $0.03 \pm 0.01$ | $90.13 \pm 0.01$ |
| | Snuba (Varma & Ré, 2018) | $95.83 \pm 0.39$ | $2.28 \pm 3.63$ | $95.89 \pm 0.30$ |
| | WARM (Ours) | $95.39 \pm 0.22$ | $1.52 \pm 0.26$ | $95.55 \pm 0.17$ |
| **Test Discriminative** | Active WeaSuL (Biegel et al., 2021) | $95.51 \pm 0.52$ | $2.88 \pm 0.52$ | $95.36 \pm 0.55$ |
| | Nashaat et al. (2018) | $92.93 \pm 0.28$ | $0.30 \pm 0.28$ | $92.71 \pm 0.29$ |
| | Snuba (Varma & Ré, 2018) | $95.51 \pm 0.37$ | $1.65 \pm 3.24$ | $95.49 \pm 0.26$ |
| | WARM (Ours) | $95.77 \pm 0.20$ | $2.21 \pm 0.19$ | $95.80 \pm 0.23$ |
| | Active Learning | $97.33 \pm 0.15$ | $47.33 \pm 0.15$ | $97.34 \pm 0.13$ |
| | FS Logistic Regression | | $-$ | |

Table 11: Performance metrics for the *Synthetic dataset*. Accuracy is final accuracy, and ΔAccuracy is change from initial to final accuracy.

| EEG Dataset | | | | |
|---|---|---|---|---|
| | **Methods** | Accuracy | ΔAccuracy | F1 |
| **Train Generative** | WARM (Ours) | $71.16 \pm 2.09$ | $5.00 \pm 1.65$ | $69.30 \pm 2.66$ |
| **Test Discriminative** | WARM (Ours) | $\mathbf{72.29 \pm 1.48}$ | $3.89 \pm 1.12$ | $\mathbf{70.19 \pm 2.14}$ |
| | Active Learning | $64.50 \pm 1.80$ | $\mathbf{14.50 \pm 1.80}$ | $62.79 \pm 16.78$ |
| | FS Logistic Regression | | $-$ | |

Table 12: Performance metrics for the *EEG dataset*. Accuracy is final accuracy, and ΔAccuracy is change from initial to final accuracy.

## A.8  WARM RANDOM SAMPLING VS UNCERTAINTY SAMPLING

| Synthetic Dataset | | | | |
|---|---|---|---|---|
| **Methods** | | Accuracy | $\Delta$Accuracy | F1 |
| **Train Generative** | WARM | $93.77 \pm 2.29$ | $0.34 \pm 2.19$ | $93.67 \pm 2.75$ |
| | WARM (US) | $95.39 \pm 0.22$ | $1.52 \pm 0.26$ | $95.55 \pm 0.17$ |
| **Test Discriminative** | WARM | $93.72 \pm 2.52$ | $0.35 \pm 2.44$ | $93.39 \pm 3.03$ |
| | WARM (US) | $95.77 \pm 0.20$ | $2.21 \pm 0.19$ | $95.80 \pm 0.23$ |

Table 13: Effect of random sampling on *Synthetic* Dataset. Accuracy is final accuracy, and $\Delta$Accuracy is change from initial to final accuracy. Accuracy and F1 score metrics are much lower for random sampling (RS) than the uncertainty sampling (US).

| Wisconsin Diagnostic Breast Cancer | | | | |
|---|---|---|---|---|
| **Methods** | | Accuracy | $\Delta$Accuracy | F1 |
| **Train Generative** | WARM (RS) | $82.86 \pm 3.80$ | $-3.82 \pm 3.63$ | $69.49 \pm 8.79$ |
| | WARM (US) | $\mathbf{93.05 \pm 0.90}$ | $\mathbf{7.78 \pm 1.25}$ | $\mathbf{90.52 \pm 1.74}$ |
| **Test Discriminative** | WARM (RS) | $83.16 \pm 3.74$ | $-4.91 \pm 3.75$ | $62.77 \pm 10.81$ |
| | WARM (US) | $91.23 \pm 0.96$ | $3.16 \pm 1.31$ | $87.27 \pm 1.19$ |

Table 14: Effect of random sampling on *Wisconsin Diagnostic Breast Cancer* Dataset. Accuracy is final accuracy, and $\Delta$Accuracy is change from initial to final accuracy. Accuracy and F1 score metrics are much lower for random sampling (RS) than the uncertainty sampling (US).

## A.9 WARM ABLATION EXPERIMENTS ON DATASET SIZE AND DOWNSTREAM MODELS

For the rest of our experiments, we created synthetic datasets using `sklearn`'s `make_classification`[5] method. The datasets comprise of clusters of points normally distributed (with standard deviation 1) about vertices of a 25-dimensional hypercube with sides of length 2, with an equal number of clusters ($= 5$) to each class. We used 25 decision tree stumps as LFs. In order to automatically create the simple LFs, we followed the same procedure outlined in Section 4.1, with the only difference being that the Random Forest classifier was trained using 250 data points sampled from the training dataset. This was done to ensure that WARM received the same amount of weak supervision regardless of the size of the dataset, and the differences in performance were solely due to the size of the dataset and not the quality of weak supervision.

| Downstream Model | Dataset size | Methods | Accuracy | ΔAccuracy | F1 |
|---|---|---|---|---|---|
| **Logistic Regression** | 1000 | Active learning | $57.70 \pm 2.01$ | $7.70 \pm 2.01$ | $62.55 \pm 4.30$ |
| | | WARM | $58.79 \pm 1.70$ | $0.36 \pm 2.30$ | $58.51 \pm 3.88$ |
| | 5000 | Active learning | $56.88 \pm 1.12$ | $6.88 \pm 1.12$ | $48.64 \pm 2.06$ |
| | | WARM | $58.80 \pm 2.89$ | $-2.96 \pm 4.99$ | $46.81 \pm 11.11$ |
| | 10000 | Active learning | $54.14 \pm 1.19$ | $4.14 \pm 1.19$ | $54.10 \pm 14.08$ |
| | | WARM | $61.20 \pm 0.98$ | $1.75 \pm 1.36$ | $67.98 \pm 1.08$ |
| | 20000 | Active learning | $62.35 \pm 1.55$ | $12.35 \pm 1.55$ | $63.86 \pm 0.76$ |
| | | WARM | $62.73 \pm 0.91$ | $0.22 \pm 1.07$ | $61.82 \pm 4.97$ |
| **Multi-Layer Perceptron** | 1000 | Active learning | $56.79 \pm 2.38$ | $6.79 \pm 2.38$ | $51.09 \pm 4.15$ |
| | | WARM | $58.67 \pm 1.87$ | $2.30 \pm 4.69$ | $57.87 \pm 4.22$ |
| | 5000 | Active learning | $60.27 \pm 1.76$ | $10.27 \pm 1.76$ | $52.24 \pm 0.59$ |
| | | WARM | $58.70 \pm 2.21$ | $-1.13 \pm 2.57$ | $44.82 \pm 10.96$ |
| | 10000 | Active learning | $55.28 \pm 2.04$ | $5.28 \pm 2.04$ | $63.70 \pm 2.60$ |
| | | WARM | $61.82 \pm 1.01$ | $1.65 \pm 1.16$ | $68.24 \pm 1.11$ |
| | 20000 | Active learning | $60.10 \pm 1.17$ | $10.10 \pm 1.17$ | $57.41 \pm 5.04$ |
| | | WARM | $62.90 \pm 0.97$ | $0.05 \pm 1.53$ | $62.11 \pm 4.87$ |
| **Random Forests** | 1000 | Active learning | $55.64 \pm 1.64$ | $5.64 \pm 1.64$ | $59.37 \pm 8.67$ |
| | | WARM | $58.91 \pm 1.42$ | $0.48 \pm 1.82$ | $59.08 \pm 4.09$ |
| | 5000 | Active learning | $59.02 \pm 1.87$ | $9.02 \pm 1.87$ | $61.98 \pm 2.09$ |
| | | WARM | $56.29 \pm 2.34$ | $-3.35 \pm 5.32$ | $41.49 \pm 12.12$ |
| | 10000 | Active learning | $55.71 \pm 2.17$ | $5.71 \pm 2.17$ | $56.82 \pm 11.39$ |
| | | WARM | $60.25 \pm 1.38$ | $1.52 \pm 1.50$ | $67.37 \pm 1.64$ |
| | 20000 | Active learning | $56.42 \pm 2.19$ | $6.42 \pm 2.19$ | $44.88 \pm 4.35$ |
| | | WARM | $62.05 \pm 1.07$ | $0.52 \pm 1.76$ | $61.24 \pm 5.66$ |

Table 15: Testing performance of different downstream models on datasets of different sizes. Accuracy is final accuracy, and ΔAccuracy is change from initial to final accuracy. Due to the complexity of the dataset, WARM outperforms Active learning on most occasions. Moreover, WARM's performance improves as the size of the dataset increases, as it is able to better estimate the accuracies of LFs.

---

[5]https://scikit-learn.org/stable/modules/generated/sklearn.datasets.make_classification.html

### A.10 WARM EXPERIMENTS ON COMPLEXITY OF DATASET

The following results are reported on a datasets with 10000 samples and varying numbers of clusters per class, created using `sklearn`'s `make_classification` method. The number of clusters per class controls the complexity of the problem, and more clusters per class is a harder classification problem than fewer clusters per class. The weak supervision sources were created in exactly the same way as mentioned in the previous section.

| Downstream Model | Complexity | Methods | Accuracy | $\Delta$Accuracy | F1 |
|---|---|---|---|---|---|
| **Logistic Regression** | 8 | Active learning | $55.45 \pm 1.34$ | $5.45 \pm 1.34$ | $62.99 \pm 9.72$ |
| | | WARM | $60.30 \pm 3.26$ | $3.70 \pm 2.96$ | $54.29 \pm 12.00$ |
| | 6 | Active learning | $61.45 \pm 4.55$ | $11.45 \pm 4.55$ | $49.31 \pm 10.97$ |
| | | WARM | $63.09 \pm 2.24$ | $-1.09 \pm 2.41$ | $55.64 \pm 8.66$ |
| | 4 | Active learning | $53.76 \pm 5.58$ | $3.76 \pm 5.58$ | $59.41 \pm 5.17$ |
| | | WARM | $59.88 \pm 1.06$ | $5.52 \pm 1.69$ | $63.93 \pm 3.02$ |
| | 2 | Active learning | $80.67 \pm 3.04$ | $30.67 \pm 3.04$ | $80.07 \pm 3.15$ |
| | | WARM | $76.73 \pm 2.18$ | $0.18 \pm 1.88$ | $76.52 \pm 4.43$ |

Table 16: Testing performance of downstream models on datasets with increasing complexity. Accuracy is final accuracy, and $\Delta$Accuracy is change from initial to final accuracy. The more clusters per class, the more complex the dataset is, as can also be seen by the deteriorating performance of the models. When the dataset is less complex, Active learning does well, but as the complexity increase WARM does better. We expect Active learning to take many more data points to learn decision boundaries in more complex datasets.

