# OpenReview forum: "ACTIVE REFINEMENT OF WEAKLY SUPERVISED MODELS"
_ICLR.cc/2022/Conference — ICLR 2022 Submitted_

### Official Review · Reviewer_379M · 2021-10-31

**Correctness:** 4
**Technical Novelty And Significance:** 3
**Empirical Novelty And Significance:** 3
**Recommendation:** 6
**Confidence:** 4

**Main Review:**

The main strengths of the paper lies in effectively combining active learning and weak supervision to generate high quality labels. The proposed approach incrementally improves the labeling functions by using the true labels obtained in each round. To achieve this they need LFs to be differentiable, which can be a drawback in some cases. Except these few changes, it is still using most of the existing weak-supervision machinery (i.e. label model, accuracy estimation and weighted combination of labeling functions to generate final labels etc.). So while the novelty might seem limited, I think it is useful in the sense that it may require minimal changes in existing weak-supervision setups to enhance it using active learning.
I don't see major problems with this work, except a few. Firstly, interpreting the accuracy results in Table 2, is slightly difficult since class distributions in the datasets are not provided i.e. are the datasets imbalanced?  From the results it looks like the cases where one can obtain good labeling functions from experts then this approach works better than active learning otherwise not. There might be cases where obtaining expert labeling functions is more costly than obtaining labels ( which is probably easier than writing LFs). It might be worthwhile to discuss these limitations in the paper.

**Summary Of The Paper:**

The paper gives a method to iteratively and interactively improve the label model in weak supervision. The approach consists of two steps, first is standard weak supervision way of weighted combination of labelling functions to generate labels. Novelty and improvement mainly comes from the second step, where true label for most uncertain data point is queried using which the parameters of the labelling functions are improved, which in turn lead to a more accurate weak supervision model. A key requirement and assumption in this paper's setup is that labelling functions are given by some learnable parameters ( i.e. they can be differentiated w.r.t. their parameters), which allows parameters updates using the true labels acquired. Empirical results on various real world datasets in medical domain show that in some cases this approach can yield a more accurate model in comparison to pure active learning approach and some recent baselines which combine weak supervision with active learning. These results also show that the paper's approach can get accuracy comparable to fully supervised model as well.

**Summary Of The Review:**

Overall, I think its a nice paper with a sound and simple approach to improve weak supervision's label quality using active learning. The contributions are novel and useful in practice. I am inclined towards accepting this paper. I need some clarifications in the experiments section to be more confident in this assessment.

---

> ### Author Response · Authors · 2021-11-22
> **Response to Reviewer 379M**
>
> We would like to thank the reviewer for their positive comments.
>
> > "*Firstly, interpreting the accuracy results in Table 2, is slightly difficult since class distributions in the datasets are not provided i.e. are the datasets imbalanced*"
>
> This is an excellent suggestion and we have added the class distribution (the proportion of positive samples) in Table 1.
>
> >"*From the results it looks like the cases where one can obtain good labeling functions from experts then this approach works better than active learning otherwise not. There might be cases where obtaining expert labeling functions is more costly than obtaining labels ( which is probably easier than writing LFs). It might be worthwhile to discuss these limitations in the paper*"
>
> This is a good point, and is in general true for data programming, which assumes that the LFs are at the very least better than random. However, we have added this as a limitation in the conclusion. We would also like to point out our experiments in Appendix A.8 which show that WARM performs well when varying amount of random noise is added to LF decision parameters.

---

### Official Review · Reviewer_aMJi · 2021-11-02

**Correctness:** 4
**Technical Novelty And Significance:** 2
**Empirical Novelty And Significance:** 2
**Recommendation:** 5
**Confidence:** 3

**Main Review:**

In equation (1), the \propto seems like the wrong relation since the left hand side is actually the softmax of the right hand side.

Why are the labels not used in estimating the labeling function accuracies?

In figure 3.(i), why is the performance not monotonic? In particular, the curve without noise peaks with 20 labels and then starts to deteriorate.

It's interesting to note that WARM's downstream performance never improves over the accuracy of the label model. It seems that a piece of the weak supervision pipeline is broken here. Perhaps using a more expressive model (random forests?) would be more appropriate for the downstream model.

I'm surprised that the active learning baseline generally outperformed the weak supervision methods even though the weak supervision methods have access to extra information (the labeling functions trained on the whole dataset).




Minor things:
I think the arguments of p_\theta are swapped throughout the paper. Look at eq (2), eq (3), and line 4 of the algorithm.


**Summary Of The Paper:**

This paper proposes an algorithm for choosing a small set of labels to improve labeling function model performance both directly and for downstream tasks. Additionally, the authors provide a general method to convert standard labeling functions to "soft" labeling functions which are differentiable with respect to some parameters (e.g. a threshold). If the labeling functions are differentiable, this paper provides a method to update the labeling function parameters. Finally, experimental results show that the method introduced outperforms other active labeling approaches for weak supervision.

**Summary Of The Review:**

The method proposed in this paper does not contain any particularly novel ideas and seems to be based on heuristics (maybe the proposed quantities could be derived from more general principles?). Additionally, it appears that weak supervision is not appropriate for the paper's empirical settings as seen by the lack of improvement from the downstream model and the stronger performance of non-weakly-supervised methods (active learning).

---

> ### Author Response · Authors · 2021-11-22
> **Response to Reviewer aMJi (1/2)**
>
> We would like to thank the reviewer for taking time to review our paper and for their comments.
>
> > "*In equation (1), the \propto seems like the wrong relation since the left hand side is actually the softmax of the right hand side.*"
>
> Thank you for pointing this out, we have adapted our notation.
>
> > "*Why are the labels not used in estimating the labeling function accuracies?*"
>
> First, we are only using a few data points (30 - 50 for all of our experiments), so there's little information that the labeled data points can add towards estimating LF accuracies. Secondly, we are indirectly incorporating the labels towards estimating LF accuracies by virtue of our algorithm which alternates between two steps. Specifically, the labels bring about changes in the LF decision parameters, which affect their votes, which then impact their accuracies.
>
> However, we agree with the reviewer that future work should investigate the impact of incorporating observed labels into the weak supervision framework.
>
> > "*In figure 3.(i), why is the performance not monotonic? In particular, the curve without noise peaks with 20 labels and then starts to deteriorate.*"
>
> We believe that the performance is not monotonic as the LFs begin to overfit to the actively sampled data points. We would like to emphasize that the curve reports the performance of the end model on the testing data. We have mentioned in the conclusion of the manuscript that future work should research strategies to prevent such overfitting.
>
> > "*It's interesting to note that WARM's downstream performance never improves over the accuracy of the label model. It seems that a piece of the weak supervision pipeline is broken here. Perhaps using a more expressive model (random forests?) would be more appropriate for the downstream model.*"
>
> The performance of the label model is measured on the training data while the end model performance is evaluated on the testing data and hence cannot be directly compared. However, based on the suggestion we have performed experiments in Appendix A.9 comparing the testing performance of different downstream models (Logistic Regression, Random Forests, and Multi-layer Perceptrons) on datasets of different sizes.
>
> > "*I'm surprised that the active learning baseline generally outperformed the weak supervision methods even though the weak supervision methods have access to extra information (the labeling functions trained on the whole dataset).*"
>
> As mentioned in the paper, we believe that active learning performs really well on simpler datasets. To show that active learning performs poorly on more complex datasets, we conducted two experiments reported Appendix A.9 and A.10. Whereas the former experiment compares the testing performance of different downstream classifiers on datasets of varying sizes, the latter experiments compares downstream model on datasets with increasing complexity. These experiments clearly show that (1)  Due to the complexity of the dataset, WARM outperforms Active learning on most occasions. (2) Moreover, WARM's performance improves as the size of the dataset increases, as it is able to better estimate the accuracies of LFs. (3) When the dataset is less complex, \texttt{Active learning} does well, but as the complexity increase WARM does better. We expect Active learning to take many more data points to learn decision boundaries in more complex datasets.
>
> > "*Minor things: I think the arguments of p_\theta are swapped throughout the paper. Look at eq (2), eq (3), and line 4 of the algorithm.*"
>
> We thank the reviewer for catching this error. We have fixed this now.

---

> > ### Comment · Reviewer_aMJi · 2021-11-22
> > **Reply to response**
> >
> > Thank you for your response.
> >
> >  - The non-monotonicity due to overfitting seems like it could be a major drawback of this method as people generally expect more data and more labels to be helpful.
> >
> >  - "The performance of the label model is measured on the training data while the end model performance is evaluated on the testing data and hence cannot be directly compared." In that case, could the performance of the label model be compared to the performance of the downstream model on the test set?
> >
> >  - In A.10, it seems there are some important details missing:
> >     how are the labeling functions created?
> >     what is the labeling budget?
> >     how many classes are used? It looks like there's 5 clusters per class.

---

> > > ### Author Response · Authors · 2021-11-22
> > > **Response to Reviewer aMJi (2/2)**
> > >
> > > > *"The non-monotonicity due to overfitting seems like it could be a major drawback of this method as people generally expect more data and more labels to be helpful."*
> > >
> > > Thank you for raising this important point. We too believe that overfitting is an important issue and future research should look at ways to mitigate it, as we have mentioned in the conclusion. To prevent overfitting to the active learning samples; we could perform early stopping, learning rate decay, gradient clipping, or cross validation to better generalize our models.
> > >
> > > > *"In that case, could the performance of the label model be compared to the performance of the downstream model on the test set?"*
> > >
> > > This is an excellent suggestion, and we are rerunning our experiments to report results from the label model on the both training and testing data. We will include these results in the camera ready version accordingly.
> > >
> > > > *"In A.10, it seems there are some important details missing: how are the labeling functions created? what is the labeling budget? how many classes are used? It looks like there's 5 clusters per class."*
> > >
> > > Thank you for pointing this out: we have added these details in Appendix A.9 and A.10. We will write them here again for convenience.
> > >
> > > **A.9:** *"For the rest of our experiments, we created synthetic datasets using sklearn's make_classification method. The datasets comprise of clusters of points normally distributed (with standard deviation 1) about vertices of a 25-dimensional hypercube with sides of length 2, with an equal number of clusters (= 5) to each class. We used 25 decision tree stumps as LFs. In order to automatically create the simple LFs, we followed the same procedure outlined in Section 4.1, with the only difference being that the Random Forest classifier was trained using 250 data points sampled from the training dataset. This was done to ensure that WARM received the same amount of weak supervision regardless of the size of the dataset, and the differences in performance were solely due to the size of the dataset and not the quality of weak supervision. "*
> > >
> > > **A.10**  *"The following results are reported on a datasets with 10000 samples and varying numbers of clusters per class, created using sklearn's make_classification method. The number of clusters per class controls the complexity of the problem, and more clusters per class is a harder classification problem than fewer clusters per class. The weak supervision sources were created in exactly the same way as mentioned in the previous section."*

---

### Official Review · Reviewer_bVTy · 2021-11-02

**Correctness:** 3
**Technical Novelty And Significance:** 3
**Empirical Novelty And Significance:** 3
**Recommendation:** 6
**Confidence:** 3

**Main Review:**

Pros:

1. The proposed method would be of great use in many real-world scenarios where labelled data is scarce (e.g. medicine).
2. To my knowledge, the authors’ proposed method (i.e., actively refining the voting weights of each labeling function and the parameters of the labeling function) is novel.
3. Generally I found the writing in the manuscript to be of high quality. As someone who is not an expert in data programming I found the manuscript easy to follow with some minor exceptions (see point 3 in “Cons”).
4. The authors experiment with their method on a variety of datasets, including one for which they use human domain experts to craft labeling functions. I greatly appreciate the application to a real-world scenario!

Cons (listed in order of importance to my score):

1. I found it difficult to assess the significance of the knowledge shift experiment results presented in Figure 2 and Figure 3 due to a lack of any results from baseline models. As such, I would appreciate it if the authors could add results from their baseline models to these Figures to (1) illustrate the severity of the knowledge shift problem and (2) (potentially) better illustrate the advantages of WARM over previous work.
2. As mentioned by the authors in Section 5, the active learning baseline outperforms WARM on half of the tested datasets. The authors say that this is due to some datasets being “simpler” than others, though I was not clear as to what “simpler” meant here. I would be willing to raise my score if the authors could provide experimental results that clearly illustrate the reasons behind the poor performance of WARM on these “simpler” datasets. For example, if “simpler” means having fewer data points, an experiment that assesses WARM's performance on simulated data with varying dataset size would be helpful to better understand when WARM may outperform existing methods vs. when it may underperform them.
3. Some of the terminology relating to the problem formulation/labeling functions is not precisely defined within the manuscript and can be confusing for readers not already familiar with data programming (e.g. I had to look outside the manuscript for a definition of the “polarity” of a labeling function).
4. The literature review in Section 2 is very data-programming specific, and does not discuss other recent approaches to active learning (e.g. [1,2,3]). Expanding the related works section would help put WARM in a broader context and assist the reviewer in assessing the significance of WARM.

[1]: “Learning Active Learning from Data” (NeurIPS 2017)
[2]: “Learning Algorithms for Active Learning” (ICML 2017)
[3]: “Active Learning with Partial Feedback” (ICLR 2019)


**Summary Of The Paper:**

This paper proposes a new method for data programming, i.e., using weak supervision to generate probabilistic training labels for unlabelled points using heuristics devised by domain experts. In particular, the authors propose WARM, a framework for iteratively improving these weakly supervised models by modifying the parameters of labeling functions and directing users to a subset of data points that, when labelled, would most improve the model.


**Summary Of The Review:**

Overall I enjoyed reading this paper. The authors’ method appears methodologically sound and it seems to provide considerable benefits when applied to complex datasets. However, there are some weaknesses in the manuscript that somewhat undermine the paper’s story. For now I am recommending a weak accept, and I am willing to revise my score upwards if the authors address my concerns.

---

> ### Author Response · Authors · 2021-11-22
> **Response to Reviewer bVTy**
>
> We thank reviewer bVTy for taking time to go through our work and for their constructive comments and pointing us to relevant work.
>
> > "*As such, I would appreciate it if the authors could add results from their baseline models to these Figures to (1) illustrate the severity of the knowledge shift problem and (2) (potentially) better illustrate the advantages of WARM over previous work.*"
>
> This is an excellent point, and we have added results from baseline models to the knowledge shift problems as well as the noise experiments in Appendix A.3. The results demonstrate that WARM either outperforms or performs on par with the baselines on both the Synthetic and Heart Disease datasets.
>
> > "*I would be willing to raise my score if the authors could provide experimental results that clearly illustrate the reasons behind the poor performance of WARM on these “simpler” datasets. For example, if “simpler” means having fewer data points, an experiment that assesses WARM's performance on simulated data with varying dataset size would be helpful to better understand when WARM may outperform existing methods vs. when it may under-perform them.*"
>
> This is an excellent suggestion. To show that active learning performs poorly on more complex datasets, we conducted two experiments reported Appendix A.9 and A.10. Whereas the former experiment compares the testing performance of different downstream classifiers on datasets of varying sizes, the latter experiments compares downstream model on datasets with increasing complexity. These experiments clearly show that (1) Due to the complexity of the dataset, WARM outperforms Active learning on most occasions. (2) Moreover, WARM's performance improves as the size of the dataset increases, as it is able to better estimate the accuracies of LFs. (3) When the dataset is less complex, \texttt{Active learning} does well, but as the complexity increase WARM does better. We expect Active learning to take many more data points to learn decision boundaries in more complex datasets.
>
> > "*Some of the terminology relating to the problem formulation/labeling functions is not precisely defined within the manuscript and can be confusing for readers not already familiar with data programming (e.g. I had to look outside the manuscript for a definition of the “polarity” of a labeling function).*"
>
> We have added clarifications to "polarity", "coverage", and other terms related to data programming in the draft. Additionally, we have made many edits to improve the clarity of our manuscript.
>
> > "*The literature review in Section 2 is very data-programming specific, and does not discuss other recent approaches to active learning (e.g. [1,2,3]). Expanding the related works section would help put WARM in a broader context and assist the reviewer in assessing the significance of WARM. ([1]: “Learning Active Learning from Data” (NeurIPS 2017) [2]: “Learning Algorithms for Active Learning” (ICML 2017) [3]: “Active Learning with Partial Feedback” (ICLR 2019))*"
>
> Thank you for pointing us to these papers. Currently, we have added these papers to the future work section in our conclusion. These works are especially interesting as they could potentially lead to further savings of human effort by using Meta Active Learning to learn query selection strategies based off of previous AL outcomes or by using partial feedback. We firmly believe that improving the active learning is vital for WARM as well as the intersection of active learning and weak supervision in general.

---

### Official Review · Reviewer_jkgK · 2021-11-02

**Correctness:** 3
**Technical Novelty And Significance:** 2
**Empirical Novelty And Significance:** 3
**Recommendation:** 5
**Confidence:** 5

**Main Review:**

Strengths:
- (S1) This paper tackles an important problem with an intuitive approach of complementing recent programmatic/weak supervision approaches with expert feedback via an active learning-style approach.
- (S2) This paper introduces a clean formulation of / argument for LFs being cast as differentiable functions, whereas to date most LFs have been non-differentiable
- (S3) The paper shows some strong results relative to recent approaches.
- (S4) The paper includes a range of datasets from synthetic (data + LFs), to 'semi-synthetic' (real data + synthetically generated LFs), to a real EEG task/dataset + LFs developed in conjunction with medical SMEs- an impressive and real world-relevant contribution.

Weaknesses:
- (W1) Simple method: While not a major drawback in isolation, it is worth nothing that the proposed approach is fairly simple and standard from a methodological/algorithmic standpoint.  From an active learning perspective: the query function is just based on model uncertainty, which is the most basic type of active learning query function (the only tweak being that it is the label model, i.e. model over LFs, but this does not change anything from an algorithm perspective).  Then, these data points are used to tune the parameters of the LFs in a manner that is also straightforward (the only tweak here being that the approach alternates between two formulations of the label model objective which is not necessarily needed... see W#1.a).
- (W1.a) The authors state that the data programming label model is not differentiable with respect to the tunable LF parameters introduced in WARM, which is not true.
- (W2) Lack of exploration of effect of differentiable LFs: Given the above lack of methodological novelty, this reviewer at least saw one of the main points of novelty and overall contribution of the paper being around the differentiable LFs themselves, and the overall setup here.  However, unfortunately this contribution was not explored in any depth (e.g. what are the tradeoffs of "softening" LFs to make them differentiable?  How should we think about this more broadly beyond the medical settings treated?  etc), which would have been interesting and significantly strengthened this aspect of the overall contribution.
- (W3) Lack of relevant ablations: In general, there were a range of ablations of the overall approach I would have thought natural- for example, what is the impact of an active learning setup vs. just using some randomly sampled labeled data to tune the LF's internal parameters?  Could these internal parameters also be learned without labeled data, following the basic Snorkel modeling approach, and how would that do?  How would the approach do without tuning the internal LF params?  Etc.
- (W4) Weak/improper comparisons: Since two of the other approaches compared to have no access to tune the internal params of the LFs and WARM does, this seems like a somewhat handicapped comparison...

**Summary Of The Paper:**

This paper proposes WARM, an active learning approach to weakly/programmatically supervised learning.  In the WARM approach, which bases off of the data programming/Snorkel paradigm for weak supervision, users write labeling functions (LFs) to programmatically label training data; these labeling functions are then modeled by the Snorkel framework for weak supervision and used to train downstream models.  In the WARM setup, these LFs are assumed to be, or cast as, differentiable.  The paper then proposes an active learning approach to sampling labeled data points to tune the parameters of these LFs, and validates this approach on several medical datasets.

**Summary Of The Review:**

Overall, the proposed approach introduces some interesting and practical ideas with some exciting experimental applications, however does not sufficiently explore the most novel elements of the contribution, ablate them sufficiently, or compare them to prior methods appropriately.

---

> ### Author Response · Authors · 2021-11-22
> **Response to Reviewer jkgK**
>
> We thank the reviewer for taking time to go through our work and for their constructive comments.
> > "*(W1) Simple method*"
>
> We agree with the reviewer that the method is simple, but firmly believe WARM's simplicity is its asset. We also agree with Reviewer 379M
> that our method's design enables users of the existing weak-supervision machinery to use active learning with minimal changes. Furthermore, we too believe that future work should investigate the impact of novel active learning strategies on the active weak supervision methodology.
>
> >"*(W1.a) The authors state that the data programming label model is not differentiable with respect to the tunable LF parameters introduced in WARM, which is not true.*"
>
> The data programming label model that WARM uses is a factor graph [1]. The factor graph uses indicator functions as factors. The LFs appear inside these factors which are noncontinuous and nondifferentiable. Thus, the probabilistic output of the label model is not differentiable with respect to the decision parameters of the labeling functions (LFs).
>
> [1] Ratner, Alexander, et al. "Training complex models with multi-task weak supervision." Proceedings of the AAAI Conference on Artificial Intelligence. Vol. 33. No. 01. 2019.
>
> > "*(W2) Lack of exploration of effect of differentiable LFs*"
>
> This is an excellent point. In practice, domain experts may prefer soft LFs over hard LFs since they are more expressive and can potentially capture the uncertainty that domain experts may have around the decision parameters. However, we agree with the reviewer, and we will make changes to discuss these ideas in the manuscript.
>
> One downside of soft LFs is that they are not general. While all soft LFs can be made crisp but not all crisp LFs can be made soft (e.g. distant supervision, text lookup LFs such as regex), and we mention this in the paper. Moreover, our current softening has an implicit slope parameter that we do not use. It is possible that in an optimal scenario for soft LFs, we may not only want to tune the scale parameter of the softening function around the threshold, but also its slope. While this may allow the softening function to encapsulate the notion of uncertainty, it may make the optimization problem non-convex.
>
> On the generalization beyond the medical settings, we would like to point out that decision stumps along with their conjuctions and disjunctions are widely used other non-medical settings [2]. Moreover, as mentioned in the paper, many frequently used classes of LFs such as multi-layer perceptrons, convolutional neural networks, neural language models, are used in other non-medical computer vision and natural language processing applications [2, 3].
>
> [2] Varma, Paroma, and Christopher Ré. "Snuba: Automating weak supervision to label training data." Proceedings of the VLDB Endowment. International Conference on Very Large Data Bases. Vol. 12. No. 3. NIH Public Access, 2018.
>
> [3] Hooper, Sarah, et al. "Cut out the annotator, keep the cutout: better segmentation with weak supervision." International Conference on Learning Representations. 2020.
>
> > "*(W3) Lack of relevant ablations*"
>
> These are really good suggestions. We have added experiments an ablation experiment comparing active learning and random sampling in Appendix A.8 of the paper on the Synthetic and Wisconsin Diagnostic Breast Cancer datasets. Our results show that the accuracy and F1 score metrics are much lower for random sampling than the uncertainty sampling (active learning).
>
> We report the improvement in performance of weak supervision without tuning any LF parameters. Specifically, in the Appendix we report both the final and $\Delta$ Accuracy, where the latter is defined as the difference in accuracy between the final label model learned using LFs optimized by WARM, and the initial label model learned via vanilla weak supervision, without tuning any LF decision parameters. We have made this information clearer in the tables.
>
> > "*(W4) Weak/improper comparisons*"
>
> We would like to emphasize that to the best of our knowledge, prior work in the intersection of active learning and weak supervision, does not tune the decision parameters of the LFs. Instead, they focus on directly improving the label model or probabilistic labels [4]. Hence, directly tuning LF decision parameters is novel to our work.
>
> [4] Biegel, Samantha, et al. "Active WeaSuL: Improving Weak Supervision with Active Learning." arXiv preprint arXiv:2104.14847 (2021).

---

### Official Review · Reviewer_JFqf · 2021-11-02

**Correctness:** 3
**Technical Novelty And Significance:** 3
**Empirical Novelty And Significance:** 3
**Recommendation:** 5
**Confidence:** 2

**Main Review:**

Strength:

The proposed method is clearly described and the writing is easy to follow.

Weakness:
1. The theoretical analysis is insufficient and I recommend the authors provide more analyses on their main contributions.
2. The general idea of active learning with weak supervision is not novel, which can be seen in [1]. The methods proposed in this work are combinations of existing ones and the authors also fail to contribute novel theoretical results.
3. The layout should be improved. I recommend the authors separate the key notations, e.g., labeling functions, from the contexts for better presentation. There are also some typos, e.g., 'dependant' should be 'dependent' in the last paragraph of page 2.

[1]. Chicheng Zhang and Kamalika Chaudhuri, Active Learning from Weak and Strong Labelers. NIPS 2015: 703-711


**Summary Of The Paper:**

In this work, the authors propose the WARM method to help conduct iterative and interactive weakly-supervised learning. Active learning is used here to refine the labeling functions by focusing on data points that are once labeled. The authors further incorporated gradient propagation to alternatively update the LF parameters and the DP model. Experimental results show that the WARM method can improve the quality of training data.

**Summary Of The Review:**

The authors should compare with more related works and provide some theoretical analyses to make this work more convincing. The ideas are totally heuristic and the experimental results are also not satisfying.

---

> ### Author Response · Authors · 2021-11-22
> **Response to Reviewer JFqf**
>
> We would like to thank Reviewer JFqf for their comments.
>
> > "*The general idea of active learning with weak supervision is not novel, which can be seen in [1]. The methods proposed in this work are combinations of existing ones and the authors also fail to contribute novel theoretical results.*"
>
> As mentioned in the paper, there has been prior work on combining active learning and weak supervision which have mostly focused on either directly improving the accuracy of the predicted labels or the label model. In our work, the novelty lies in WARM's ability to change the decision parameters of the LFs to improve the label model and its predictions directly. We thank the reviewer for pointing out the interesting paper, which assumes access to both a weak yet cheap, and a strong yet expensive labeler, and uses active learning to minimize queries to the strong labeler. While related, our setting is different since we have access to multiple weak labelers, along with a strong labeler, and use active learning to improve the accuracy of the weak labelers.
>
> > "*The layout should be improved. I recommend the authors separate the key notations, e.g., labeling functions, from the contexts for better presentation. There are also some typos, e.g., 'dependant' should be 'dependent' in the last paragraph of page 2.*"
>
> We have fixed the typographical errors, and are actively working to improve the layout of the paper for the final camera ready version. In this process, we have already made several changes to improve the clarity of the paper.

---

### Author Response · Authors · 2021-11-22
**General Response**

We would like to thank all of the reviewers for their timely and constructive reviews and suggestions. We were encouraged to note that reviewer 379M found our contributions to be novel and useful in practice, and reviewer bVTy found our methodology sound.

We have made several edits to the paper to improve clarity as well as conducted multiple experiments based off of reviewer suggestions. Additionally, we have provided individual responses to each reviewer below, as well as an updated PDF (the reviewers can click the "Show Revisions" link to see differences compared to the first draft).

Additional Experiments:
1. **Ablation Experiments:** As mentioned in the paper, we believe that active learning performs really well on simpler datasets. To show that active learning performs poorly on more complex datasets, we conducted two experiments reported in Appendix A.9 and A.10. Whereas the former experiment compares the testing performance of different downstream classifiers on datasets of varying sizes, the latter set of experiments compares the downstream model on datasets with increasing complexity. These experiments clearly show that (1) Due to the complexity of the dataset, WARM outperforms Active learning on most occasions. (2) Moreover, WARM's performance improves as the size of the dataset increases, as it is able to better estimate the accuracies of LFs. (3) When the dataset is less complex, active learning does well, but as the complexity increases, WARM does better. We expect Active learning to take many more data points to learn decision boundaries in more complex datasets.

2. **Active vs Random Sampling:** We have added an ablation experiment comparing active learning and random sampling in Appendix A.8 of the paper on the Synthetic and Wisconsin Diagnostic Breast Cancer datasets. Our results show that the accuracy and F1 score metrics are much lower for random sampling than the uncertainty sampling (active learning).

3. **Baselines on Knowledge Shift Experiments:** We have added results from baseline models to the knowledge shift problems as well as the noise experiments in Appendix A.3. The results demonstrate that WARM either outperforms or performs on par with the baselines on both the Synthetic and Heart Disease datasets.

4. **Baselines on Noise Experiments:** While the reviewers did not specifically request this, we have also added baselines results to the noise experiments in Appendix A.4. We find that noisy LFs hurt both label and end model performance. However, WARM can iteratively de-noise LFs by actively collecting a few more labeled examples. Even with 75% noise initially, WARM can refine LFs such that the resulting label and end model perform on par with those trained using LFs with no noise. WARM also consistently outperforms other baselines.

---

### Decision · Program_Chairs · 2022-01-20

**Decision:**

Reject

**Comment:**

The authors propose WARM, a novel method that actively queries a small set of true labels to improve the label function in weak supervision. In particular, the authors propose a methodology that converts the label function to "soft" versions that are differentiable, which are in term learnable with true labels using proper updates of parameters. Empirical results on several real-world data sets demonstrate that the method yields a pretty strong performance.

The reviewers generally agree that the idea of making the labeling functions differentiable is conceptually interesting. They are also positive about the simplicity and the promising performance. They share joint concerns on whether the idea has been sufficiently studied in terms of the design choices and completeness of the experiments. For instance, the authors can conduct deeper exploration of the trade-off for differentiable LFs. They can also study active learning strategies that are beyond basic uncertainty sampling. While the authors have provided more studies about those exploration and ablation studies during the rebuttal, generally the results are not sufficient to convince most of the reviewers. In future revisions, the authors are encouraged to clarify its position with respect to existing works that combine active learning and weakly-supervised learning.

The authors position the paper as more empirical than theoretical. So the suggestion from some reviewers about more theoretical study is viewed as nice-to-have but not a must.